# Genome-Wide Identification of ATP-Binding Cassette (ABC) Transporter Gene Family and Their Expression Analysis in Response to Anthocyanin Transportation in the Fruit Peel of Eggplant (*Solanum melongena* L.)

**DOI:** 10.3390/ijms26167848

**Published:** 2025-08-14

**Authors:** Hesbon Ochieng Obel, Xiaohui Zhou, Songyu Liu, Liwei Xing, Yan Yang, Jun Liu, Yong Zhuang

**Affiliations:** 1Institute of Vegetable Crops, Jiangsu Academy of Agricultural Sciences, Nanjing 210014, China; 20230024@jaas.ac.cn (H.O.O.); 20100029@jaas.ac.cn (X.Z.); 20180052@jaas.ac.cn (S.L.); 20170007@jaas.ac.cn (Y.Y.); 20110024@jaas.ac.cn (J.L.); 2Jiangsu Key Laboratory for Horticultural Crop Genetic Improvement, Nanjing 210014, China; 3College of Horticulture, Ludong University, Yantai 264025, China; xlw18vvvvv@m.ldu.edu.cn

**Keywords:** eggplant, ABC gene family, anthocyanin, fruit peel color

## Abstract

The ATP-binding cassette (ABC) gene family represents one of the most extensive and evolutionarily conserved groups of proteins, characterized by ATP-dependent transporters that mediate the movement of substrates across cellular membranes. Despite their well-documented functions in various biological processes, the specific contributions of ABC transporters in eggplant (*Solanum melongena* L.) remain unexplored. To address this gap, we conducted a comprehensive genome-wide identification and expression profiling of ABC transporter-encoding genes in eggplant. Our investigation identified 159 *SmABC* genes encoding ABC transporter that were irregularly dispersed across all 12 chromosomes. The encoded proteins exhibited considerable diversity in size, with amino acid lengths varying from 55 to 2628 residues, molecular weights ranging between 4.04 and 286.42 kDa, and isoelectric points spanning from 4.89 to 11.62. Phylogenetic analysis classified the *SmABC* transporters into eight distinct subfamilies, with the ABCG subfamily being the most predominant. Subcellular localization predictions revealed that most SmABC proteins were localized to the plasma membrane. Members within the same subfamily exhibited conserved motif arrangements and exon–intron structures, suggesting functional and evolutionary conservation. Promoter analysis identified both shared and unique *cis*-regulatory elements associated with transcriptional regulation. We identified 9 tandem duplication gene pairs and 20 segmental duplication pairs in the *SmABC* gene family, with segmental duplication being the major mode of expansion. Non-synonymous to synonymous substitutions (Ka/Ks) analysis revealed that paralogs of *SmABC* family genes underwent mainly purifying selection during the evolutionary process. Comparative genomic analysis demonstrated collinearity between eggplant, *Arabidopsis thaliana*, and tomato (*Solanum lycopersicum*), confirming homology among *SmABC*, *AtABC*, and *SlABC* genes. Tissue-specific expression profiling revealed differential *SmABC* expression patterns, with three distinct genes, *SmABCA16*, *SmABCA17* and *SmABCG15*, showing preferential expression in purple-peeled fruits (A1, A3, and A5 accessions), implicating their potential involvement in anthocyanin transport. Functional validation via *SmABCA16* silencing led to a significant downregulation of *SmABCA16* and reduced purple coloration, indicating its regulatory role in anthocyanin transport in eggplant fruit peel. This comprehensive genomic and functional characterization of ABC transporters in eggplant establishes a critical foundation for understanding their biological roles and supports targeted breeding strategies to enhance fruit quality traits.

## 1. Introduction

ATP-binding cassette (ABC) transporters represent one of the most extensive and evolutionarily conserved protein superfamilies, ubiquitously present across animal, plant, and microbial species [1]. Structurally, ABC transporters feature two highly conserved domains: a nucleotide-binding domain (NBD) and a transmembrane domain (TMD) typically composed of five to six α-helices [2]. These proteins are further subdivided into three architectural classes: (i) full-molecule transporters containing dual NBDs and TMDs, (ii) half-molecule transporters possessing a single NBD-TMD pair, and (iii) soluble proteins that lack TMDs entirely [3,4]. The TMD serves as the substrate-recognition module, enabling selective membrane translocation, while the NBD harbors characteristic conserved motifs—Walker A, Walker B, ABC signature, H-loop, and Q-loop—that mediate ATP binding and hydrolysis [5,6]. Notably, the NBD is positioned intracellularly and contains the invariant Walker A/B sequences and the LSGGQ motif, exhibiting lower conservation.

In plants, comprehensive genomic analyses have revealed that ABC transporters are systematically classified into eight distinct subfamilies (ABCA-ABCF and ABCI), with the notable absence of the ABCH subfamily that is present in other organisms [7,8]. The ABCB, ABCC, and ABCG subfamilies have received particular research attention due to their diverse functional roles across species [9]. Plant ABC transporters can be further categorized based on their structural organization. The full-transporter class comprises ABCA (a homolog of ABC1), ABCB/C (also known as multidrug resistance protein), and ABCG (pleiotropic drug resistance protein) subfamilies, which contain complete transporter domains. In contrast, the half-transporter group comprises ABCA (a homolog of ABC2), ABCB, a transporter of the mitochondrion and a TAP/transporter associated with antigen processing, ABCD associated with peroxisomal membrane protein 3, and ABCG associated with white-brown complex protein subfamilies, which possess only partial transporter domains. The ABCE (RLI/RNase L inhibitor), ABCF (general control non-repressible), and ABCI (structural maintenance of chromosomes and NAP/non-intrinsic ABC proteins) subfamilies represent soluble ABC transporters that lack transmembrane domains [10,11,12].

The ABCA subfamily exhibits structural diversity, comprising both full-transporter (AOH/ABC1 homolog) and half-transporter (ATH/ABC2 homolog) variants. *Arabidopsis thaliana*, for instance, encodes one AOH-type and sixteen ATH-type ABCA proteins [13]. Phylogenetic analyses indicate that AOH-type genes appear exclusively in dicotyledonous species (such as Arabidopsis, *Vitis vinifera*, and *Gossypium* spp.) while being conspicuously absent from monocot genomes such as *Zea mays* [14,15,16]. The ABCB subfamily encompasses three functionally distinct classes: MDR (multidrug resistance), ATM (mitochondrial ABC transporter), and TAP (antigen processing) proteins. Arabidopsis contains 29 ABCB members [13], which have been functionally characterized to play crucial roles in auxin homeostasis regulation and transport [17,18,19,20], ion translocation, and heavy metal detoxification [21,22].

The ABCC subfamily encodes multidrug resistance-associated proteins (MRPs) that play pivotal roles in plant metabolite transport and detoxification processes [23,24]. Functional studies have demonstrated that ABCC members mediate diverse transport activities: *AtABCC4* facilitates folate transport in Arabidopsis [25], while *VvABCC1* in grapevine (*Vitis vinifera*) and *AtABCC2* participate in anthocyanin sequestration [26]. Similar functions have been reported for *FaABCC8* in strawberry (*Fragaria × ananassa*) [27] and *PpABCC1* in peach (*Prunus persica*) [28]. In cereals, *TaABCC13* (a homolog of maize LPA1) mediates glutathione-conjugated transport in wheat (*Triticum aestivum*) [29], while *ZmMRPs* in maize and *OsMRP15* in rice (*Oryza sativa*) facilitate vacuolar accumulation of flavonoids and anthocyanins [30,31]. Notably, Arabidopsis *AtABCC1* and *AtABCC14* transport acylated anthocyanins through a glutathione-independent mechanism, distinct from the conjugation-dependent transport of anthocyanin 3-O-glucosides [32]. Beyond secondary metabolite transport, ABCC members like *AtMRP1* and *AtMRP2* confer heavy metal resistance [33,34]. The ABCD subfamily comprises peroxisomal membrane proteins (PMPs), with *AtPMP2* regulating seed germination [35,36]. The ABCG subfamily is the most extensive group and includes pleiotropic drug resistance (PDR) and white-brown complex (WBC) proteins that are involved in transporting terpenoids, alkaloids, lipids, and volatile compounds [37,38]. In contrast, the ABCE, ABCF, and ABCI subfamilies comprise soluble proteins that lack transmembrane domains and transport capabilities [7].

Eggplant (*Solanum melongena* L.), a globally significant solanaceous crop, holds substantial agricultural importance as a widely cultivated vegetable. Within the Solanaceae family, it ranks as the third most economically valuable crop, surpassed only by potato (*Solanum tuberosum* L.) and tomato. Purple eggplant is vital for its richness in anthocyanin content in the fruit peel. Eggplant anthocyanins are important in the provision of health benefits such as antioxidants, anti-inflammatories, and potential anticancer properties. Anthocyanin plays a commercial role in enhancing fruit color for consumer preference and marketability, and it is used as natural food colorants [39]. While specific ABC genes from different species have been reported, for example, *AtABCC1/2* and *AtABCC14* in Arabidopsis [26,32] and *PpABCC1* in peach [28], little or no information is available regarding the role of ABC genes in anthocyanin transport in solanaceous crops like eggplant. The initial genome assembly of eggplant was first reported in 2014 [40]. Subsequent advances in sequencing technologies have facilitated the development of progressively enhanced genome assemblies [41,42]. While these high-quality genomic resources have enabled the comprehensive identification of numerous gene families in eggplant, systematic characterization of the ATP-binding cassette (ABC) transporter gene family remains unexplored. This knowledge gap significantly hinders functional investigations of ABC genes in this economically important crop. Of particular interest is the anthocyanin-rich purple-fruited eggplant, which has gained considerable commercial popularity due to its enhanced nutritional and purported medicinal properties. Therefore, understanding the potential involvement of ABC transporters in regulating fruit peel pigmentation and anthocyanin accumulation represents a research priority with both scientific and agricultural significance.

Studying the ABC gene family in eggplant is vital to gaining insights into their characteristics. Therefore, we conducted a genome-wide analysis with protein sequences obtained from the eggplant genome (http://47.92.172.28:12068/Eggplant/browse/SME, accessed on 20 January 2025). This investigation characterized the *SmABC* gene family by assessing their molecular properties, genomic positions, structural organization, conserved domains, phylogenetic relationships, promoter-regulatory elements, and syntenic patterns. Furthermore, the study evaluated the expression dynamics of *SmABC* genes across eggplant fruit varieties with distinct peel pigmentation to elucidate their biological functions. Prior RNA-seq work (PRINA868105) identified differentially expressed genes (DEGs) associated with pigmentation, including structural and regulatory anthocyanin biosynthesis genes (such as *CHS*, *UFGT*) and general metabolism. The current research reveals how *ABC* transporters could be a possible mediator in vacuolar sequestration, a bottleneck in peel coloration. Silencing specific *ABC* (*SmABCA16*) reduced peel coloration, thus transforming correlative RNA-seq observations into causal evidence. These findings provide valuable guidance for future research on the functional roles of *SmABC* genes in eggplant. Additionally, they establish a fundamental basis for improving fruit peel pigmentation-related characteristics in breeding initiatives aimed at enhancing crop quality.

## 2. Results

### 2.1. Identification and Physicochemical Analysis of the SmABC Gene Family

In total, 196 ABC candidate protein sequences were identified in eggplant, and their sequence information was provided in Appendix A. Among these sequences, 37 encoded <200 amino acid residues, which is most likely to be pseudogenes or not full-length ABC transporters due to genome annotation errors (Appendix A). These short sequences were excluded from further analysis. These short sequences included 1 gene from ABCA, 10 from ABCB, 6 from ABCC, 1 from ABCE, 17 from ABCG, and 2 from the ABCI subfamily. Thus, a total of 159 sequences were chosen for further analysis. To verify the reliability of these BLAST in the eggplant genome database (http://eggplant-hq.cn/Eggplant/home/index, accessed on 20 January 2025) results, the 159 protein sequences were subjected to Pfam analysis. The 159 protein sequences were confirmed to have conserved domains or the nucleotide-binding domain (NBD) and TMD (transmembrane domain) regions and motifs of ABC proteins (Appendix A). As a result, a total of 159 genes encoding for ABC transporters were identified in the eggplant genome (Appendix A).

The 159 ABC proteins were grouped into 8 subfamilies, including 31 ABCAs, 31 ABCBs, 22 ABCCs, 1ABCD, 2 ABCEs, 8 ABCFs, 51 ABCGs, and 13 ABCIs. The genes were systematically designated and assigned numerical identifiers based on their specific chromosomal positions within each subgroup (Appendix A). Substantial diversity was observed among the ABC protein sequences, showing variations from 55 to 2628 amino acids in length and 6.02 to 286.42 kDa in molecular mass. The encoded ABC proteins displayed wide ranging physical characteristics, with amino acid chain lengths varying 47-fold (55–2628 aa) and molecular weights differing by nearly 280 kDa (6.02–286.42 kDa). Furthermore, the theoretical isoelectric points (pI) of the ABC proteins exhibited considerable variation, ranging from 4.04 to 11.62, with ABCs with pI > 7 and pI < 7 being considered basic proteins and acidic proteins, respectively. The instability index ranged from 22.97 to 67. Instability indices less than 40 indicated stable proteins, while those with an instability index greater than 40 were unstable. The aliphatic index ranged from 49.92 to 22.97. Most of the SmABC proteins (60%) were hydrophobic proteins (grand average of hydropathicity >0), while 40% were hydrophilic proteins (grand average of hydropathicity <0). Bioinformatics analysis predicted ABC protein localization in diverse cellular structures, spanning membrane systems (plasma membrane, ER, Golgi), organelles (chloroplasts, mitochondria, vacuoles), and soluble compartments (cytoplasm, cytosol, nucleus).

### 2.2. Phylogenetic Analysis of the SmABC Gene Family

Phylogenetic analysis of *SmABC* family members was constructed using the neighbor-joining method (Appendix A). Using SmABC (159) and AtABC (131) member protein sequences, a phylogenetic tree was constructed to compare the homology of eggplant SmABC and Arabidopsis AtABC protein sequences. Using AtABC proteins as phylogenetic markers, we identified eight SmABC subfamilies (ABCA-ABCI, excluding ABCH) (Figure 1). The conserved homology patterns between species imply shared ancestral origins and probable conservation of physiological roles among these ABC transporters.

### 2.3. Chromosomal Distribution, Collinearity, and Gene Duplication Analysis of SmABC in Eggplant

We determined the physical distribution of *SmABC* genes across eggplant chromosomes through comprehensive genome annotation analysis (Figure 2). The eggplant gene chromosomal localization analysis revealed that the *SmABC* genes are distributed across all 12 chromosomes. The *SmABC* genes were evenly dispersed throughout eggplant chromosomes without apparent distribution bias. A significant cluster was observed on chromosome 3, accounting for 25 genes, followed by chromosome 6, which has 24 genes. Chromosomes 2, 10, and 12 have an equal number of genes (12 genes), while chromosomes 4 and 9 have the lowest number of *SmABC* genes, comprising 9 and 11 genes, respectively. However, two ABC genes (*SmABCG67* and *SmABCG68*) were anchored on the scaffold. Moreover, the eggplant genome has substantial variation in SmABC gene counts across chromosomes, indicating extensive gene loss and duplication throughout evolutionary history. This genomic pattern provides a foundation for reconstructing the evolutionary diversification of ABC transporters in eggplant.

The syntenic paralog pairs of *SmABC* genes were identified within the eggplant genome. We identified 20 pairs of segmental duplicate genes and 9 pairs of tandem duplicate genes in the *SmABC* family members (Figure 3A, Appendix A). Notably, the *SmABCA* subfamily has 7 pairs of segmental duplicate genes and 1 pair of tandem duplicate genes. Subfamilies, *SmABCB* and *SmABCG*, each possess 4 segmental and 2 tandem gene duplications. The subfamily SmABC has 1 segmental and 3 tandem gene duplications, while subfamily *SmABCF* and *SmABCI* each have 2 segmental duplications with the absence of a tandem duplication gene pair. The 20 pairs of segmental duplicate genes and 9 pairs of tandem duplicates were distributed on all 7 chromosomes except chromosome 10 (Figure 3A). Moreover, the duplicated paralog *SmABC* transporter pairs belong to the same subfamily. Further, we identified synonymous (Ks) and non-synonymous (Ka) values to explore the selective pressures on these paralog pairs to understand the expansion of this gene family in eggplant (Appendix A). The Ka/Ks of eggplant *SmABC* tandem duplicated gene pairs ranged from 0.179 to 0.734, with a mean value of 0.322. The Ka/Ks of segmental duplicated gene pairs ranged from 0.093 to 0.466, with a mean value of 0.251. The Ka/Ks values of all tandem duplicated and segmental duplicated *SmABC* gene pairs were less than one, which implies that these genes evolved under the influence of purifying selection. The average Ka/Ks value of tandem duplication genes (0.322) was higher than that of segmental duplication genes (0.251), indicating that tandem duplication evolved faster than other duplication events. The duplication time of eggplant ABC paralog pairs was estimated using a relative Ks measure as a proxy for time, spanning from 1.056 to 167.99 million years ago (MYA), with an average duplication time of 75 MYA.

To elucidate the evolutionary relationships of *SmABC* genes, we performed comparative synteny analysis of ABC transporter genes among eggplant, Arabidopsis, and tomato (Figure 3B). The result showed that eggplant has 50 and 113 collinear gene pairs with Arabidopsis and tomato, respectively (Appendix A). In Arabidopsis and tomato, some conserved syntenic relationships between *SmABC* genes and multiple ABC family members across species imply strong selective pressure maintaining these genomic arrangements, underscoring their critical evolutionary functions. In contrast, a one-to-one collinear gene pairs between the species indicate a high degree of conservation.

### 2.4. Gene Structure and Conserved Motif Analysis of SmABC in Eggplant

Like most plant genes, SmABC family members exhibit variable intron–exon architectures (1–42 exons per gene) (Appendix A). This structural diversity enables comparative analyses of gene evolution, particularly for understanding lineage-specific expansion patterns in Solanaceae ABC transporters. We characterized 12 significantly conserved protein motifs (Motif 1–12; Figure 4A–F) in SmABC family members. Comprehensive motif profiling revealed substantial variation in conserved domains across SmABC proteins (0–12 motifs per gene). Nine core motifs (1–3, 5–8, 10, and 12) demonstrated particularly high conservation, being present in >85% of family members, suggesting their fundamental role in transporter structure and function. The motif sequence information is provided in Appendix A. The co-evolution of protein motifs and gene structures within phylogenetic subfamilies indicates lineage-specific preservation of functional domains. Structural constraints maintain transporter functionality and the ancestral origin of core ABC architectures. These patterns likely reflect strong purifying selection on essential transporter conformations. Our integrated analysis reveals that motif conservation patterns precisely mirror phylogenetic boundaries and that subfamily-specific motif configurations correspond to differential substrate binding affinities, varied cellular localization signals, and distinct regulatory mechanisms. These structural specializations probably underlie the functional diversification of eggplant ABC transporters.

### 2.5. Analysis of Cis-Acting Elements in the SmABC Gene Family

In plants, the gene promoter is a critical *cis*-acting regulatory element located upstream of the coding region, governing transcriptional initiation. Our investigation of cis-regulatory elements revealed the presence of distinct motifs within the promoter sequences of *SmABC* (Figure 5A,B). These *cis*-regulatory elements could be broadly divided into eight categories: light-responsive elements, promoter-related elements, hormone-responsive elements, environmental stress-related elements, development-related elements, site-binding-related elements, and other elements. The light-responsive category comprised the highest proportion of cis-regulatory elements, with 30 identified motifs, including 3-AF1 binding site, 4cl-CmA1b, AAAC motif, ACA motif, ACE, AE box, Aas-1, ATI-motif, ATC-motif, ATCT-motif, box4, CAG motif, chs-CMA19, chs-CMA29, chs-CMA2b, chs-unit, GA- motif, GAP-box, GATA motif, G-box, GTI-box, LAMP element, TCCC-motif, L-box, LS7, Sbp-CmA1C, SP1, TCT, and F-box. Additionally, twelve hormone-responsive elements were detected, spanning nearly all major phytohormone pathways. These included ABRE (abscisic acid response), ERE (ethylene response), GARE-motif and P-box (gibberellin response), SARE and TCA-element (salicylic acid response), TGA-element and AuxRR-core (auxin response), as well as CGTCA-motif, TATC-box, and TGACG-motif (MeJA response). The third category included 10 stress-responsive *cis*-regulatory elements associated with environmental challenges, such as heat stress (Box-W1, Box-W3), fungal elicitor response, drought (MBS, a MYB binding site), low-temperature adaptation (LTR), dehydration, low temperature and salt stress (DRE), anoxic-specific inducibility (GC), anaerobic induction (ARE), and wound response (W-box, box-s and WUN). The subsequent category comprised 15 elements linked to plant growth and development, including AACA-motif, AAGAA-motif, AC-I, AC-II, meristem expression (CAT-box), CCAAT-motif, O2-site (Zein metabolism), as1, circadian, endosperm expression (GCN4-motif), seed-specific regulation (RY-element), Skn-1_motif, pallisade mesophyll cell (HD-Zip1, HD-Zip2), flavonoid biosynthetic gene regulation (MBS1), and MSA-like. Finally, two additional categories were identified: promoter-related elements (8 types) and site-binding-related elements (7 types) (Appendix A). Generally, the number of *cis*-regulatory elements varied among the SmABC genes, with SmABCG34/35 containing a single *cis*-regulatory element consisting of DRE and Myb binding site, respectively.

### 2.6. Expression of SmABC Genes on Eggplant Fruits with Different Peel Colors

Contemporary RNA-seq methodologies, including single-cell and long-read sequencing platforms, have revolutionized our capacity to deconvolute complex transcriptional landscapes, revealing spatiotemporal gene regulation patterns with unprecedented resolution. Here we investigated the evolutionary divergence of ABC gene family expression patterns in eggplant through comparative transcriptomic analysis and their potential role to anthocyanin transport on fruit peel. Transcriptome sequencing data of five eggplant germplasms with different peel colors were analyzed. The study incorporated five eggplant germplasms exhibiting distinct peel pigmentation labeled as dark-purple peel—A1; green peel—A2; black-purple with green calyx—A3; white peel cultivar—A4; and cultivars of reddish-purple—A5. Expression profiling of *SmABC* genes was conducted through analysis of RNA-seq datasets generated in a recent study [43]. Similarly, the anthocyanin quantification of the germplasm linked to this current study is outlined in a previous study [43]. ABC family genes have diverse expression patterns among the different fruit peel colors of eggplant, thus suggesting different functions of the ABC gene on fruit peel coloration (Figure 6A). The FPKM values are provided in Appendix A.

In the dark and red purple cultivars (A1), three *SmABC* genes, including *SmABCA16*, *SmABCA17*, and *SmABCG15*, showed high expression affinity and green and white fruit peels showed low expression. These genes were upregulated on the purple (A1/A3) and reddish (A5) eggplant varieties, and these differential expression profiles suggest that specific *SmABC* transporters may participate in anthocyanin transport in eggplant peel. Moreover, the correlation analysis between *SmABC* genes and anthocyanin structural genes, as well as anthocyanin content, showed various variations (Figure 6B). The total anthocyanin content (mg/g values) of the different genotypes is referenced from our recent research group data [43]. Moreover, structural genes *SmCHI*, *SmDFR*, *SmANS* and *SmUFGT*, as well as the total anthocyanin content, have a positive correlation (>0.8) with *SmABCA16*, *SmABCA17*, and *SmABCG15*. The results depict the potential role of these *SmABC* genes in response to anthocyanin transport on the peel of eggplant fruits.

To explore the expression and potential function of the *SmABC* family in eggplant fruit development and peel pigmentation, qRT-PCR was applied to analyze the expression of 20 *SmABC* genes (Figure 7). From the results, the selected *SmABC* genes displayed diverse expression trends. Similar to RNA-seq data, *SmABCA16*, *SmABC17*, and *SmABCG15* showed high expression in the purple/red color eggplant peels compared to the green peel fruits, with *SmABCA16* showing the highest expression. Similarly, qRT-PCR of other subgroups B-G on the fruit peel corroborates the RNA-seq results, thus further signifying the validity of the RNA-seq data. The tissue samples for qRT-PCR were obtained from the dark-purple peel (A1) germplasm. The expression of *SmABC* varied among the other fruit tissues (flesh, calyx, root, stem, and leaves). The variation in expression among the different tissues suggests tissue-specific expression and functional roles of ABC genes in eggplant fruits. The *SmABC* gene exhibiting high expression levels in the purple eggplant and, subsequently, a decreasing expression in green peel fruits, particularly *SmABCA16*, is suggested to have a potential regulatory role in the transport of anthocyanin in eggplant fruits. Therefore, further functional study warrants the analysis of *SmABCA16* to elucidate its role in purple coloration and anthocyanin transport and accumulation.

### 2.7. Analysis of the Effects of Silencing SmABCA16 on Fruit Color and the Expression Pattern

In this study, three genes (*SmABCA16*, *SmABC17*, and *SmABCG15*) were initially selected for VIGS screening, but only *SmABCA16* was reported since it produced a clear, observable phenotype, resulting in altered peel color, thus making it biologically relevant for further study. The other two genes did not show detectable phenotypic changes upon silencing, making them less interesting for reporting (Appendix A). Based on the expression patterns, the eggplant *SmABCA16* may contribute to anthocyanin transport on the peel of the eggplant fruit. The silenced fruits (TRV2-*SmABCA16*) showed decreased purple color with more greenish/white patches around the injected region compared to the control (TRV2) (Figure 8A). Quantitative qRT-PCR analysis confirmed successful suppression of *SmABCA16* expression in VIGS-treated eggplant fruits, demonstrating significant transcriptional downregulation (*p* < 0.05) compared to control plants (Figure 8B). The result implies successful silencing of *SmABCA16* and its potential role in anthocyanin transport in the peel of purple eggplant fruits.

## 3. Discussion

Ubiquitous in nature, ABC proteins comprise one of the largest known families of molecular transporters with extensive functional diversity across taxa. ABC genes transport many substrates, such as secondary metabolites, phytohormones and heavy metal detoxification [24]. Genomic and biochemical studies have revealed the presence and functional significance of ABC-type transporters in various plant systems, demonstrating both conserved functions and lineage-specific expansions. Genome-wide analyses have identified 129 members in *Arabidopsis thaliana* [44], 123 in *Oryza sativa* [45], 133 in *Zea mays* [46], 117 in *Prunus dulcis* [4], 320 in *Gossypium hirsutum* [15], 261 in *Glycine max* [47], and 170 in *Camellia sinensis* [48]. Notably, Solanaceae species exhibit significant diversity, with *Solanum tuberosum* harboring 54 members [49], *Solanum lycopersicum* containing 154 [50,51], and *Capsicum annuum* possessing approximately 200 ABC transporters [52]. These findings highlight the dynamic evolution of this gene family across angiosperms, with variations in family size potentially reflecting adaptations to ecological niches or physiological demands. Our genome-scale analysis revealed 159 *SmABC* transporter genes in eggplant, organized into 8 phylogenetically distinct subfamilies with non-random chromosomal distribution. Functional characterization of this expanded gene family suggests that evolutionary diversification has generated substantial biochemical variation relevant to anthocyanin transport mechanisms. The largest subfamily members were ABCG, ABCA, and ABCC, while ABCF, ABCE, and ABCD were the least abundant. The ABCG subfamily is one of the most prominent gene families in plants and, also, is the largest in the eggplant ABC gene family, accounting for 34.69% of all ABC genes. Many proteins belonging to this subfamily have been found to play a crucial role in conferring resistance to pathogens, antibacterial terpenes, and herbicides. Furthermore, these transporters likely participate in the translocation of signaling compounds and emission of volatile metabolites [53,54], mirroring the functional patterns observed in related species, including tomato [51] and pepper [52].

Phylogenetic analysis revealed substantial structural divergence within the SmABC transporter family proteins, reflecting their evolutionary diversification. Comprehensive sequence characterization demonstrated marked variations in physicochemical properties, including protein length (ranging from 55 to 2628 amino acids), molecular weight (4.04–286.42 kDa), and theoretical isoelectric points (4.89–11.62). We further observed considerable differences in charge distribution (positively charged residues vs. negatively charged residues), hydrophobicity (GRAVY indices), aliphatic character, and predicted stability (instability indices). These variations suggest functional specialization among SmABC members, potentially influencing substrate specificity, subcellular localization, and interaction partners. Notably, the observed heterogeneity aligns with the functional diversity of ABC transporters in other plant systems, supporting their roles in metabolite transport, stress adaptation, and secondary metabolism regulation. ABC transporters are predominantly localized to membrane-bound organelles in plant cells, including the plasma membrane, mitochondrial envelope, and tonoplast. Our subcellular localization predictions identified 106 SmABC proteins targeted to the plasma membrane, a distribution pattern consistent with characterized ABC family members in Malus domestica [55] and Citrus sinensis [48]. Chromosomal mapping revealed an asymmetric distribution of ABC transporter genes across eggplant’s 12 chromosomes, with clustering patterns suggesting lineage-specific duplication events as the primary driver of this genomic arrangement [56]. Diversity in gene structure organization, particularly exon–intron arrangements, contributes significantly to the functional diversification of gene families and provides valuable phylogenetic insights [57]. Structural analysis revealed that closely related ABC transporters maintain similar genomic architectures, with canonical full-length members typically comprising more than 1200 amino acid residues [44]. Genomic characterization of the 159 SmABC transporters revealed striking structural variation, with protein lengths spanning nearly 50-fold (55–2628 aa). Gene architecture analysis showed exon numbers ranging from 1 to 34, mirroring observations in Arabidopsis [44] and strawberry [58] ABC families. Notably, the ABCG subfamily contained the largest proteins (>2000 aa), while ABCIs were the most compact exon-rich genes (>20 exons), predominantly belonging to full-size transporters. Gene duplication analysis revealed that multiple copies of genes in a gene family could have evolved due to the flexibility provided by events of whole-genome tandem and segmental duplications. Gene duplication, segmental or tandem, has been documented in several plant gene families [59]. The Ka/Ks ratios of paralog pairs were <1, representing purifying selection in these SmABC paralogs at the protein level. This finding corroborates other gene families in plants, such as BURP in Medicago and ACD in tomato, which contain a few or even no paralog pairs undergoing positive selection [60]. In our study, ABC paralog pairs underwent purifying selection. Purifying selection, also called negative selection, is a type of natural selection that removes harmful or deleterious mutations from a population, preserving the functional integrity of genes. Its implications are significant in evolutionary biology and genetics [61]. Twelve signature motifs were identified across subfamilies (Motif 1–12). Conserved motif patterns suggest maintained substrate specificity in related members. Length variation correlates with domain loss/expansion events during evolution. These findings align with comparative analyses in Solanaceae (such as tomato ABCCs) [51], supporting conserved structural evolution despite sequence divergence. To delineate the evolutionary history and genomic organization of *SmABC* transporters, we conducted comparative synteny analysis across the eggplant genome, identifying conserved gene linkages and duplication events. There were 50 and 113 pairs of collinear genes between eggplant versus Arabidopsis, and between eggplant versus tomato, respectively. Specifically, the relationship between eggplant and Arabidopsis and tomato suggests partial evolutionary discrepancy, potentially maintaining conserved functional roles. Promoter analysis revealed abundant *cis*-regulatory elements that interact with transcription factors to modulate expression patterns. These regulatory motifs mediate precise transcriptional responses to environmental stimuli and tissue-specific demands, as characterized in prior studies [62]. The presence of stress-related *cis*-elements and hormone-response motifs in *SmABC* promoters indicates these transporters likely coordinate eggplant’s defense against diverse environmental challenges, with pronounced involvement in abiotic stress adaptation. In addition, *SmABC* genes contain MYB-related binding sites involved in drought inducibility. MYB proteins are also known to be involved in anthocyanin biosynthesis in plants [63].

Expression profiling of *SmABC* family genes in eggplant revealed distinct tissue-specific patterns, with notable variations observed across roots, stems, leaves, calyx, and fruit tissues (peel and flesh). Notably, several *SmABC* genes exhibited significantly higher expression levels in leaves compared to fruits and other tissues. This preferential expression in leaves suggests potential functional specialization, possibly linked to their physiological roles in photosynthetic and stress-responsive processes. The divergent expression profiles between leaves and fruit peel may arise from tissue-specific transcriptional regulation and functional diversification. In many purple-fruited species, leaves often exhibit elevated expression of anthocyanin-related genes, which are crucial for UV photoprotection and mitigation of oxidative stress [64]. Conversely, fruit peel-specific gene expression is often modulated by developmental signals and environmental factors associated with ripening [65]. Furthermore, variations in metabolic demands between source (leaves) and sink (fruit peel) tissues, as well as resource partitioning strategies, may further shape these expression dynamics [66].

Expression profiling across eggplant peel colors and other tissues provides critical insights into the dynamic regulation and molecular functions of key genes. *SmABC* genes exhibited distinct preferential expression patterns, particularly in purple (A1, A3) and red (A5) peel varieties, suggesting their potential involvement in anthocyanin biosynthesis (Figure 6A). Among these, three *SmABC* genes showed marked upregulation in pigmented peels, further supporting their putative role in anthocyanin-related pathways. qRT-PCR validation confirmed the RNA-seq findings, with consistent expression trends observed across different fruit peels. These results establish a foundational framework for identifying and characterizing functionally significant genes in eggplant. Anthocyanin biosynthesis occurs on the cytosolic face of the endoplasmic reticulum, followed by compartmentalization into vacuoles for storage, a process potentially mediated by ABC transporters and other regulatory factors.

While anthocyanin biosynthesis has been extensively studied, intracellular transport remains a critical yet poorly characterized rate-limiting step in anthocyanin accumulation. Recent advances have identified several tonoplast-localized transporters, including MATE and ABC-type proteins, as key players in anthocyanin sequestration. Among these, ABCC/MRP-type transporters have been functionally validated across multiple species, including *PpABCC1* in peach (*Prunus persica*; [28], *AtABCC1/2/14* in Arabidopsis [26,32], *MdABCI17* in apple (*Malus domestica*; [67], and *FvABCC8* in strawberry (*Fragaria vesca*; [27]. In contrast, ABCA subfamily members such as *AtABCA9* and *AtABCA10* have been primarily associated with lipid transport [68,69], with their potential role in anthocyanin trafficking remaining largely unexplored in solanaceous crops. To address this gap, we investigated the function of *SmABCA16* in eggplant using virus-induced gene silencing (VIGS). Silencing *SmABCA16* reduced its expression and purple coloration, confirming its regulatory role in anthocyanin transport within the fruit peel. These findings provide the first experimental evidence implicating an ABCA transporter in anthocyanin accumulation, offering new insights into the mechanistic basis of pigmentation in solanaceous crops. Our study identifies *SmABCA16* as a critical transporter involved in anthocyanin accumulation in eggplant, providing the first functional evidence of an ABC transporter regulating pigment deposition in Solanaceae crops. This finding expands the known roles of ABC transporters beyond model plants and staple crops, highlighting their conserved yet specialized functions in secondary metabolite transport. Since anthocyanins confer both nutritional and aesthetic value, *SmABCA16* could serve as a molecular marker for breeding high-pigment eggplant varieties. The differential expression of *SmABCA16* in fruit peel versus flesh suggests its utility in engineering anthocyanin distribution for novel striped or fully pigmented varieties. Orthologs of *SmABCA16* in tomato and pepper could be explored to enhance anthocyanin content, addressing consumer demand for antioxidant-rich produce. Beyond Solanaceae, structural and functional parallels with ABC transporters in Arabidopsis and fruits like apple suggest that manipulating similar genes could optimize pigment profiles in diverse crops.

## 4. Materials and Methods

### 4.1. Genome-Wide Identification and Protein Characterization Analysis of ABC in Eggplant

For comprehensive identification of ABC transporter genes in the eggplant genome, we initiated the analysis by accessing the high-resolution reference genome [42] from the Zhejiang Academy of Agricultural Sciences genomic repository (available at: http://eggplant-hq.cn/Eggplant/home/index, accessed on 20 January 2025). Using *Arabidopsis thaliana* ABC protein sequences as query templates, we conducted a systematic homology search against the eggplant proteome through local BLASTP alignment, applying a stringent E-value cutoff of 1 × 10^−5^ to ensure identification of significant matches. Additionally, we employed the keywords “ABC” to search the eggplant genome database and screened the results to identify potential candidates as members of the eggplant ABC. The short protein sequences less than 200 amino acids were filtered out as they are likely to be pseudogenes or not full-length ABC transporters due to genome annotation errors. The remaining protein sequences were subsequently submitted to the Pfam database (http://pfam.xfam.org/, accessed on 25 January 2025) and SMART website (http://smart.embl-heidelberg.de/, accessed on 25 January 2025) for verification of the integrity of their ABC domains (ABC transporter domain (PF00005), ABC transporter transmembrane region domain (PF00664), ABC 3 transport family (PF00950), and ABC 3 transport family (PF00950). After removing redundant sequences, the remaining candidates were assigned numbers and names based on their homology and phylogenetic relationship with the Arabidopsis ABC. Finally, using the online tool ProtParam provided by the ExPASy website (https://web.expasy.org/protparam/, accessed on 25 January 2025), we analyzed the physical and chemical properties of the identified eggplant ABC. Finally, subcellular localization prediction was conducted using WoLF PSORT (https://wolfpsort.hgc.jp/, accessed on 25 January 2025).

### 4.2. Phylogenetic Analysis of ABC Genes in Eggplant

To understand the phylogenetic relationship of ABC proteins between eggplant and Arabidopsis, all the identified ABC amino acid sequences of eggplant and Arabidopsis were used to construct the phylogenetic tree. The SmABC and AtABC protein, multiple sequence alignment was performed using MEGA X software and the alignment was done using MUSCLE with default parameters. The phylogenetic tree was constructed using the neighbor-joining method with 1000 bootstrap replicates with the default settings in MEGA X software [70] and saved in newick format for further modification. Decoration of the phylogeny was accomplished using iTOL (https://itol.embl.de/itol.cgi/, accessed on 27 January 2025) [71].

### 4.3. Gene Structure Conserved Motif Analysis of ABC in Eggplant

Information on ABC gene exons and introns was extracted from the eggplant whole-genome annotation file, and visualization was carried out using TBtools-II version 2.326 [72]. The conserved motif of the ABC proteins was determined using the multiple expectation maximization for motif elicitation (MEME) suite (http://meme-suite.org/tools/meme, accessed on 27 January 2025), an online program for motif discovery [73], and visualized using TBtools. Using the MEME suite (Version 5.4.1), the motifs were searched with these parameters: with default parameters, the ‘number of motifs’ set to 12. The MEME result output was visualized using TBtools.

### 4.4. Chromosome Localization, Collinearity, and Gene Duplication Analysis of ABC in Eggplant

The location information of ABC genes in the annotated genome file of eggplant was obtained, and TBtools software was used to locate them on the 12 chromosomes. Subsequently, we obtained the genome and annotation files of *Arabidopsis thaliana* from the Ensemble plants website (https://plants.ensembl.org/index.html, accessed on 27 January 2025), as well as the genome and annotation files of *Solanum lycopersicum* from the Solanaceae Genomics Network website (https://solgenomics.net/, accessed on 27 January 2025). The Circos tool was used for drawing chromosome collinearity diagrams. The Ka/Ks Calculator tool was used to calculate Ka (non-synonymous substitution rate), Ks (synonymous substitution rate), and Ka/Ks values for duplicated genes. The occurrence of gene duplication events in terms of million years was computed following the formulae: T = Ks/2λ × 10^−6^, where λ = 1.5 × 10^−8^ synonymous substitutions in each site per annum for dicotyledons [74]. TBtools-II (Ver 2.326) was used to analyze the collinearity relationships between species and to draw collinearity relationship diagrams for the comparative Arabidopsis–eggplant and tomato–eggplant groups.

### 4.5. Identification of Cis-Acting Elements of ABC in Eggplant

*Cis*-acting element annotation upstream 2000 bp sequences of the SmABC genes were extracted as the promoter sequences using TBtools. The promoter sequences were submitted to PlantCare (http://bioinformatics.psb.ugent.be/webtools/plantcare/html/, accessed on 20 February 2025) to predict potential *cis*-acting elements. The *cis*-acting elements were then classified and statistically analyzed according to their functional annotations. Finally, TBtools-II was used to visualize the distribution of *cis*-acting elements.

### 4.6. Expression Analysis Using Transcriptomic Data

The expression data of eggplant *SmABC* genes among eggplant varieties of different peel colors were obtained from our recent research group RNA-seq data (PRINA868105) [43]. The eggplant varieties of different peel color phenotypes that were used for RNA-seq analysis comprised the following: A1 (Suqie 6 cultivar-black-purple with purple calyx), A2 (Suqie 801 cultivar green), A3 (Bulita cultivar-black purple with green calyx), A4 (Suqie 11 cultivar-white), and A5 (Hangqie 1 cultivar-reddish). The A1 cultivar is a Chinese variety with elongated fruits and purple calyces, in contrast to A3, which is a European type with ovate-shaped fruits and green calyces. Meanwhile, A2, A4, and A5 are also Chinese cultivars, displaying green, white, and reddish-purple fruit peels, respectively. Total RNA was extracted from the fruit peels of the five varieties using the RNeasy Plant Mini Kit (Qiagen, Hilden, Germany). RNA concentration, integrity, and purity were measured using a Nanodrop and Agilent Bioanalyzer 2100 (Agilent Technologies, Santa Clara, CA, USA). The libraries were sequenced on an Illumina HiSeq platform (San Diego, CA, USA), generating 6.38–7.45 Gb of clean reads per sample after filtering. These reads were aligned to the eggplant reference genome (http://eggplant-hq.cn, accessed on 21 June 2021), with mapping rates ranging from 96.64% to 97.91%. Gene expression levels were normalized using FPKM (fragments per kilobase per million mapped reads), which accounts for transcript length and total mapped reads. The samples were grown in a greenhouse at the Luhe experimental station of the Jiangsu Academy of Agricultural Sciences. Fruit peel samples were collected 25 days post-anthesis. The growth conditions and sampling of fruit tissues are well illustrated in the previous report [43].

### 4.7. Quantitative Real-Time PCR Analysis of SmABC Genes

Total RNA was isolated using the RNA Plant Extraction Kit (Tiangen, Beijing, China), followed by cDNA synthesis with a reverse transcription kit (TaKaRa, Shiga, Japan) according to the manufacturer’s instructions. Specific primers for *SmABC* were designed based on the coding sequences using Primer Premier 5. The qPCR mixture, with a total volume of 20 μL, consisted of 10 μL of 2 × SYBR Green qPCR Master Mix (Vazyme, Nanjing, China), 7 μL of double-distilled H_2_O (10 pM), and 1 μL each of the forward and reverse primers, 1 μL of cDNA template (100 ng). The qRT-PCR assays were conducted using the Light Cycler^®^ 96 system by Roche (Rotkreuz, Switzerland), with the thermal cycling conditions set to an initial denaturation at 95 °C for 2 min, followed by 40 cycles of 95 °C for 30 s, 55 °C for 30 s, and 72 °C for 1 min. Fluorescence signals were recorded during the extension phase of each cycle. Finally, a melting curve analysis was performed to verify the specificity of the amplified products. Data analysis used the 2^−(ΔΔCt)^ method, with the Actin gene as the internal normalization reference gene. All experiments were conducted with three biological and technical replicates. All primers used for qRT-PCR are listed in Appendix A.

### 4.8. Vector Construction and Transformation of Eggplant

To investigate the functional role of *SmABCA16*, we performed virus-induced gene silencing (VIGS) using a tobacco rattle virus (TRV)-based system (TRV1/TRV2), following established protocols [75]. A partial open reading frame (ORF) of *SmABCA16* was amplified via PCR and cloned into the TRV2 vector to generate the TRV-*SmABCA16* construct. The recombinant plasmid was introduced into Agrobacterium tumefaciens strain GV3101, which was then cultured to an OD600 of 0.8. Bacterial cells were pelleted and resuspended in infiltration buffer (10 mM MgCl2, 10 mM MES [pH 5.5], and 150 µM acetosyringone), followed by induction at 25 °C for 3 h.

For VIGS infiltration, the TRV2-*SmABCA16* suspension was mixed with TRV1 at a 1:1 (*v*/*v*) ratio and injected into the calyx region of developing purple eggplant fruits using a microsyringe without a needle. The calyx was excised immediately after infiltration to enhance systemic silencing. Treated fruits were kept in darkness at 21 °C for 24 h and subsequently transferred to standard growth conditions (25 °C, 16 h light/8 h dark photoperiod) for seven days. Silencing efficiency and target gene expression levels were then assessed. Fruits displaying successful suppression of *SmABCA16* were selected for further analysis, and transcript levels were quantified via qRT-PCR using primers listed in Appendix A.

### 4.9. Statistical Analysis

All quantitative experiments were performed in triplicate to ensure reproducibility and statistical reliability. Data processing and preliminary analyses were conducted using Microsoft Excel 2019. For statistical evaluation, we performed Student’s *t*-tests and/or Tukey’s test (*p* < 0.01 threshold for significance) using SAS version 9.4 (SAS Institute Inc., Cary, NC, USA).

## 5. Conclusions

In this study, a comprehensive genome-wide analysis identified 159 ABC transporter genes in eggplant, distributed unevenly across its 12 chromosomes. Phylogenetic classification revealed that these SmABC proteins cluster into eight distinct subfamilies. Consistent with their phylogenetic relationships, motif and conserved domain analyses demonstrated structural conservation within each subfamily, implying functional similarities among members of the same subgroup. *Cis*-regulatory element analysis predicted that all *SmABC* genes harbor light-responsive motifs, suggesting a potential link between their expression and light-mediated regulation. Additionally, the presence of multiple hormone-responsive elements indicates that *SmABC* genes may be governed by a complex signaling network involved in various physiological processes. Expression profiling during fruit development revealed that certain *SmABC* genes are likely involved in regulating key metabolic pathways, including anthocyanin biosynthesis and fruit color initiation. To functionally validate these findings, virus-induced gene silencing (VIGS) was employed to suppress *SmABCA16* expression in purple-skinned eggplant fruits. This resulted in a significant reduction in purple pigmentation. qRT-PCR confirmed that *SmABCA16* silencing correlated with decreased expression of both the target gene and anthocyanin-related pigmentation. It is important to note that, in this study, we prioritized *SmABCA16* for functional validation via silencing due to its strong association with expression in purple colored fruit and the feasibility of testing its role within the scope of our experimental framework. However, further additional validation approaches, such as overexpression, CRISPR knockout, or complementation assays, will be of the essence for a follow-up study. These findings presented in this study provide valuable insights into the functional roles of *SmABC* genes in anthocyanin regulation and offer a foundation for future research to improve eggplant fruit quality and other crops.

## Figures and Tables

**Figure 1 ijms-26-07848-f001:**
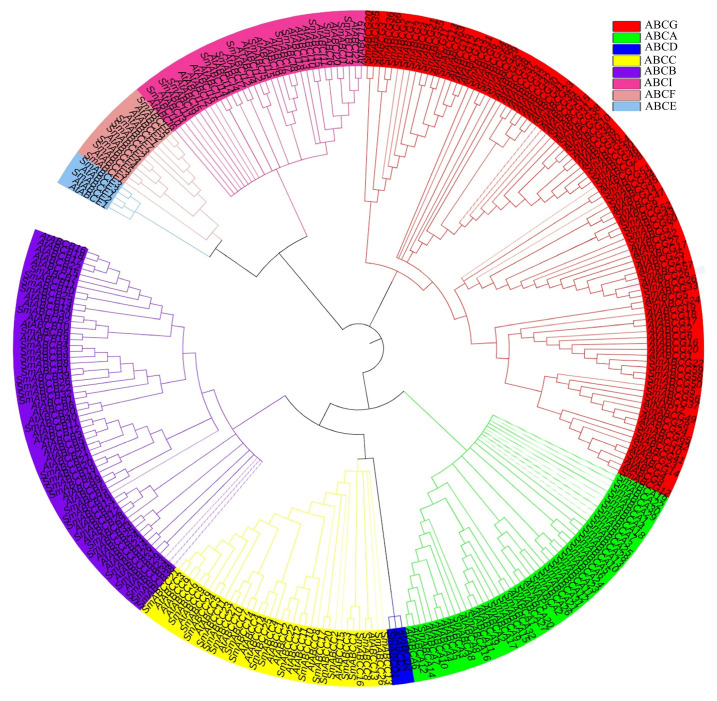
Phylogenetic analysis of ABC genes in eggplant and Arabidopsis. The ABC group is categorized into eight groups (ABCA, ABCB, ABCC, ABCD, ABCE, ABCF, ABCG, and ABCI, marked by different colors.

**Figure 2 ijms-26-07848-f002:**
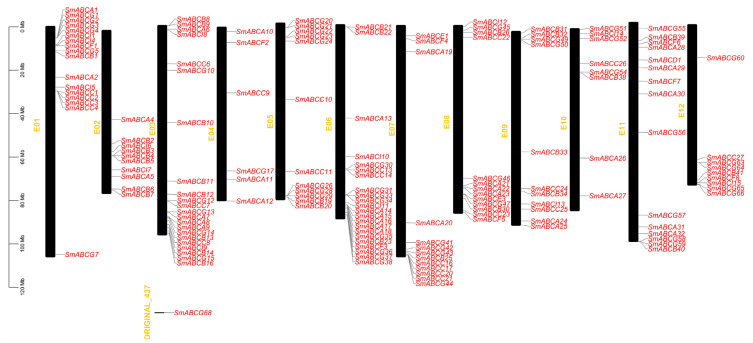
Chromosomal locations of *SmABC* genes in eggplant. Chromosomal localization was visualized through TBtools. The chromosome number is indicated in yellow to the left of each chromosome. The scale bar describes the relative lengths of the chromosomes in megabases (Mb).

**Figure 3 ijms-26-07848-f003:**
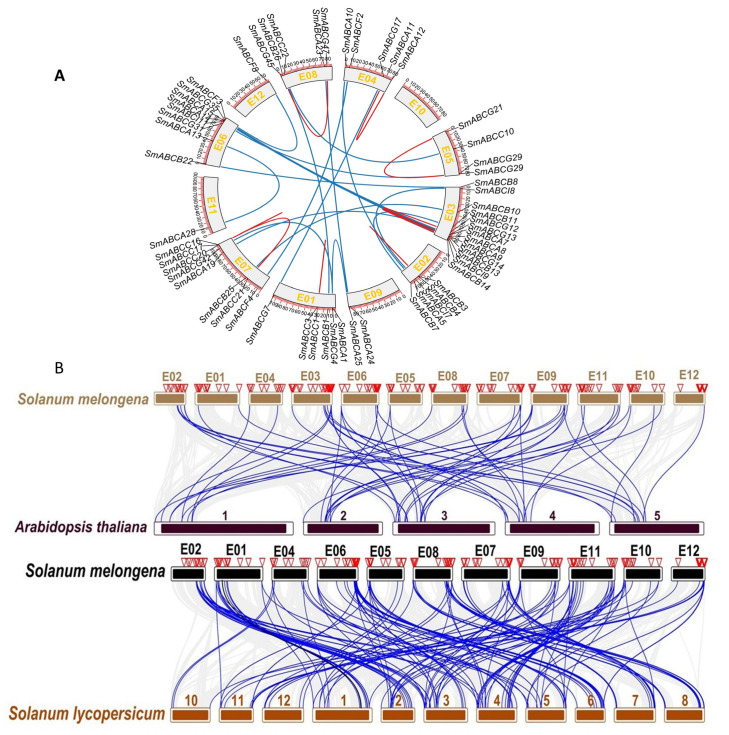
Synteny analysis of *SmABC* genes. (**A**) Synteny analysis of *SmABC* genes within eggplant. The blue line represents segment replication, while the red line shows tandem replication between gene family members. (**B**) Synteny analysis of eggplant ABC genes with ABC family members from Arabidopsis and tomato. Each horizontal line represents a chromosome with a chromosome number indicated above the horizontal line. The blue lines indicate the collinear gene pairs, while the gray lines indicate non-orthologous gene pairs.

**Figure 4 ijms-26-07848-f004:**
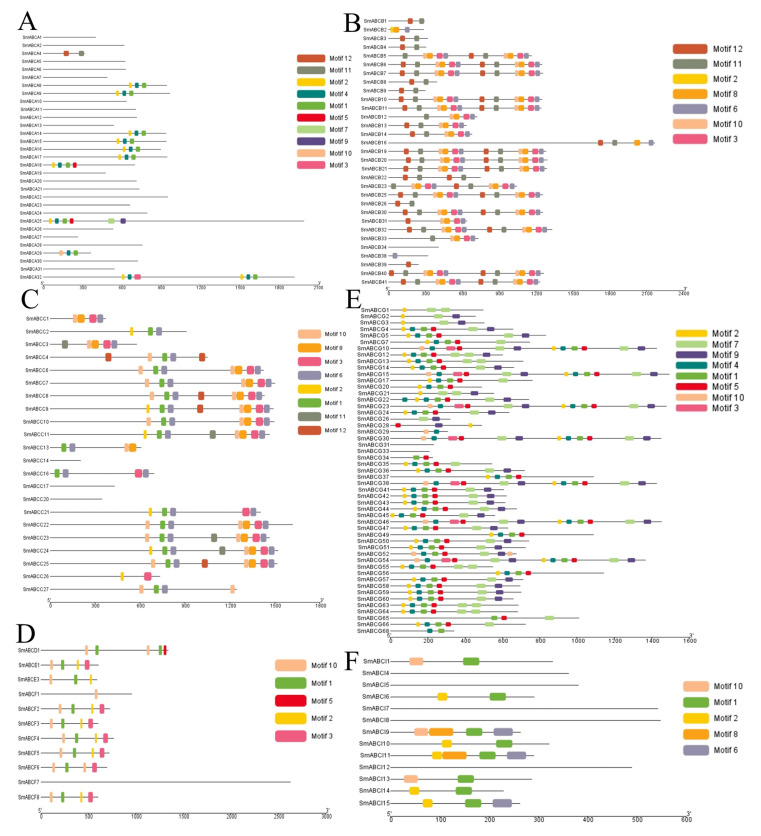
Conserved motifs of SmABC proteins in eggplant. The conserved motifs were elucidated using MEME with complete protein sequences. Different motifs are represented by different colors, numbered 1–12. The black lines represent the non-conserved sequences. (**A**) (ABCA subfamily); (**B**) (ABCB subfamily); (**C**) (ABCC subfamily); (**D**) (ABCD, E and F subfamily); (**E**) (ABCG subfamily) and (**F**) (ABCI subfamily).

**Figure 5 ijms-26-07848-f005:**
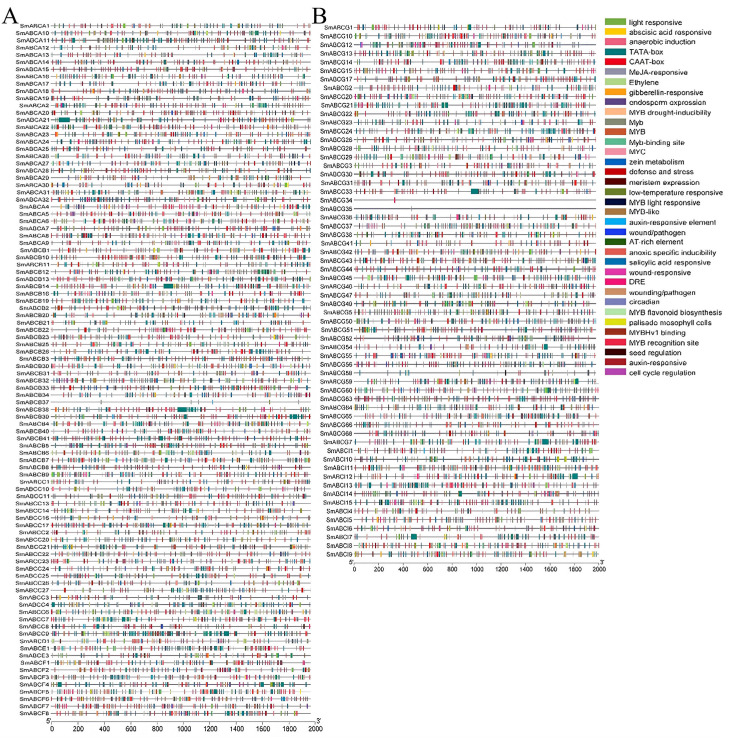
*Cis*-element analysis of the promoter of eggplant ABC family genes. (**A**) The *cis*-elements for ABCA, ABCB, ABCC, ABCD, ABCE, and ABCF. (**B**) The *cis*-elements for ABCG and ABCI. The type of each *cis*-element is indicated by colors for each *SmABC* gene.

**Figure 6 ijms-26-07848-f006:**
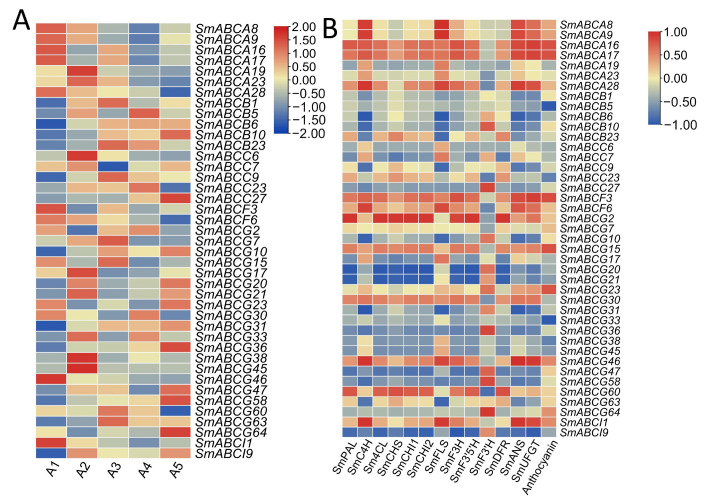
Expression profile of *SmABC* genes on the peel of different eggplant varieties. (**A**) Quantified expression of *SmABC* transporters using FPKM values from RNA-seq data visualized as Log_2_ (FPKM+1) transformed expression across five phenotypically distinct cultivars. The legend red color shows upregulated genes, while blue indicates downregulated genes, with the pale color representing non-significant expression of the genes per eggplant germplasm (**B**) Pairwise correlation analysis (Pearson, *p* < 0.05) between *SmABC* gene expression profiles, anthocyanin content (mg/g FW) and key anthocyanin biosynthetic genes. The legend red color shows a positive correlation coefficient, blue shows a negative correlation coefficient, and pale color shows non-significant correlation coefficient value.

**Figure 7 ijms-26-07848-f007:**
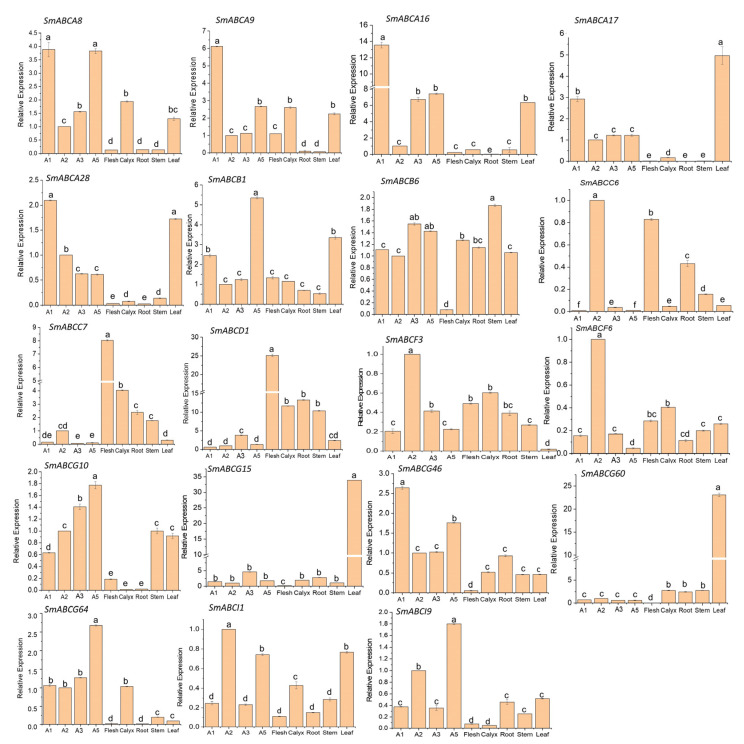
qRT-PCR expression pattern analysis of *SmABC* genes in fruit and plant tissues. Expression pattern on the fruit peel (dark-purple peel—A1; green peel—A2; black-purple with green calyx—A3; reddish-purple—A5) roots, stem, calyx, and leaves. Statistical significance was determined through one-way analysis of variance (ANOVA), followed by Tukey’s honestly significant difference (HSD) post hoc test (α = 0.01). Mean separations are indicated by superscript letters, while vertical bars represent ±1 standard deviation of the mean.

**Figure 8 ijms-26-07848-f008:**
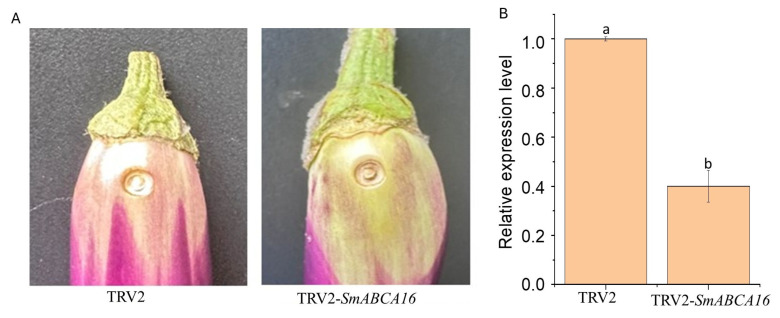
Virus-induced gene silencing of *SmABCA16*. (**A**) Fruit phenotype of TRV2 and TRV2-*SmABCA16*, 10 days post-injection. (**B**) The expression level of *SmABCA16* after silencing. Data were analyzed using one-way ANOVA with Tukey’s HSD multiple comparisons (family-wise error rate = 0.01). Dissimilar letter annotations denote statistically distinct means (*p* < 0.01), and error bars indicate sample variability.

## Data Availability

All needed genome sequences and genome annotation files of eggplant were obtained from the eggplant genome ((http://47.92.172.28:12068/Eggplant/browse/SME, accessed on 20 January 2025), and the published ABC sequences of *Arabidopsis thaliana* were acquired from the TAIR database (http://www.arabidopsis.org/, accessed on 20 January 2025) and tomato genome database (https://solgenomics.net/, accessed on 20 January 2025). The transcriptome sequencing data of different eggplant fruit peel tissues used in this study were obtained from a previous report RNA-Seq datasets presented and can be found in the NCBI Sequence Read Archive (SRA) under the accession number PRINA868105 (https://www.ncbi.nlm.nih.gov/bioproject/?term=PRINA868105, accessed on 20 January 2025). All databases in this study are available to the public.

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
