# Peer review of "Genome-Wide Identification of ATP-Binding Cassette (ABC) Transporter Gene Family and Their Expression Analysis in Response to Anthocyanin Transportation in the Fruit Peel of Eggplant (Solanum melongena L.)"

_ijms, 2025, doi:10.3390/ijms26167848_

Round 1
Reviewer 1 Report
Comments and Suggestions for Authors
This study provides valuable insights into the ABC transporter family in eggplant and its role in anthocyanin transport. Addressing the following points—particularly the validation of true ABC genes, comprehensive reporting of VIGS data, and correction of analytical inconsistencies—is essential to strengthen the manuscript's rigor, accuracy, and interpretability prior to further consideration for publication.
1. The ABC transporter identification protocol requires stricter validation to exclude pseudogenes. Several identified "genes" (e.g., SmABCG25 [55 aa], SmABCB18 [62 aa], SmABCG67 [62 aa]) encode proteins significantly shorter than the minimum length required for functional NBD (~200 aa) or TMD (~250 aa) domains. Inclusion of such truncated sequences likely represents pseudogenes and risks misleading downstream analyses. Recommendation: Implement structural validation using Pfam/SMART to confirm the presence of essential NBD and TMD domains. Explicitly filter out sequences lacking these complete domains. Reference methodology: Sánchez-Fernández et al. (DOI: 10.1007/s004250100661).
2. The significance label "dc" for SmABCG6 expression in roots (Figure 7) is erroneous.
3. The manuscript states that three genes (SmABCA16, SmABCA17, and SmABCG15) were selected for VIGS screening (Line 307). However, only results for SmABCA16 are presented. Recommendation: Provide phenotypic and expression data for all three genes tested in the VIGS screen. Negative results (no phenotype for SmABCA17 and SmABCG15) are scientifically valuable and must be included, at minimum within the Supplementary Materials.
4. Capsicum annuum (pepper) genome data is cited for synteny analysis (Line 489). This is inconsistent with the stated comparison species (Arabidopsis and tomato).
5. Latin scientific names (e.g., Solanum melongena, Arabidopsis thaliana) are not consistently italicized throughout the text (e.g., Line 191).
6. In Figure 7, some gene names are written in regular script, while others are italicized.
Author Response
Thank you very much for taking your valuable time to review our manuscript. we have vigorously revised the manuscript and improved the contents. Specific details based on the comments are attached. Thank you very much, and we are looking forward to a positive response.
Comments and Suggestions for Authors
This study provides valuable insights into the ABC transporter family in eggplant and its role in anthocyanin transport. Addressing the following points—particularly the validation of true ABC genes, comprehensive reporting of VIGS data, and correction of analytical inconsistencies—is essential to strengthen the manuscript's rigor, accuracy, and interpretability prior to further consideration for publication.
- The ABC transporter identification protocol requires stricter validation to exclude pseudogenes. Several identified "genes" (e.g., SmABCG25 [55 aa], SmABCB18 [62 aa], SmABCG67 [62 aa]) encode proteins significantly shorter than the minimum length required for functional NBD (~200 aa) or TMD (~250 aa) domains. Inclusion of such truncated sequences likely represents pseudogenes and risks misleading downstream analyses. Recommendation: Implement structural validation using Pfam/SMART to confirm the presence of essential NBD and TMD domains. Explicitly filter out sequences lacking these complete domains. Reference methodology: Sánchez-Fernández et al. (DOI: 10.1007/s004250100661).
Response: Thank you very much for this comment.We have revised and found 37 protein sequences encoded <200 amino acid residues, which is most likely to be pseudogenes or not full-length ABC transporters due to genome annotation errors (Supplemental Table S2). Hence we have excluded from further analysis. These short sequences included 1 gene from ABCA, 10 from ABCB, 6 from ABCC, 1 from ABCE, 17 from ABCG and 2 from the ABCI subfamily. Thus, a total of 159 sequences were chosen for further analysis. To verify the reliability of these BLAST results, the 159 protein sequences were subjected to Pfam analysis. The 159 protein sequences were confirmed to have conserved domains either or the Nucleotide-Binding Domain (NBD) and TMD (Transmembrane Domain) regions and motifs of ABC proteins as outlined in Supplemental Table S3.
The significance label "dc" for SmABCG6 expression in roots (Figure 7) is erroneous.
Response: We have corrected it accordingly. Thank you
3. The manuscript states that three genes (SmABCA16, SmABCA17, and SmABCG15) were selected for VIGS screening (Line 307). However, only results for SmABCA16 are presented. Recommendation: Provide phenotypic and expression data for all three genes tested in the VIGS screen. Negative results (no phenotype for SmABCA17 and SmABCG15) are scientifically valuable and must be included, at minimum within the Supplementary Materials.
Response: Thank you for this important suggestion. We have provided the phenotype information for SmABCA17 and SmABCG15 as supplementary Figure S3.
4. Capsicum annuum (pepper) genome data is cited for synteny analysis (Line 489). This is inconsistent with the stated comparison species (Arabidopsis and tomato).
Response: Thank you for noting this mistake. We have corrected it accordingly to Solanum lycopersicum.
5. Latin scientific names (e.g., Solanum melongena, Arabidopsis thaliana) are not consistently italicized throughout the text (e.g., Line 191).
Response: We have checked the entire manuscript and uniformly italicized where applicable.
6. In Figure 7, some gene names are written in regular script, while others are italicized.
Response: Thank you for noting this. We have uniformly italicized the gene names in figure 7.
We appreciate your valuable time for reviewing our manuscript and highlighting critical areas that need adjustment. We have now revised it accordingly, and we believe it quality has improved. We will be grateful for positive feedback. Thank you so much.
Reviewer 2 Report
Comments and Suggestions for Authors
This manuscript presents a valuable genome-wide analysis of ABC transporters in eggplant, with a focus on anthocyanin transport mechanisms. The identification of 196 SmABC genes and functional validation of SmABCA16 represent significant contributions. However, several issues regarding clarity, data presentation, and contextual depth require revision:
#1 Abstract contains excessive methodological/background details .
#2 Intruduction lacks critical background on: (a) nutritional/commercial value of eggplant anthocyanins; (b) ABC transporter roles in anthocyanin sequestration;(c) the knowledge gap in Solanaceae ABC research.
#3 Redundant parentheses need removal in line 68.
#4 Disorganized punctuation requires correction for sentence coherence in lines 90-91.
#5 Low image clarity impedes motif interpretation, and motif-free proteins (e.g., SmABC15) lack biological explanation in Figure 4.
#6 Unidentified purple blocks in SmABCG34/38 ( B) need labeling, and overall resolution requires enhancement in Figure 6.
#7 Misplaced symbols or sentence structure requires verification/correction in line205.
#8 Accessions A1-A5 lack peel color descriptions and anthocyanin quantification data (Lines 254-255).
#9 Color scaling contradicts numerical values , necessitating standardized visualization in Figure 6A.
#10 Values exceed the expected [-1,1] range, and anthocyanin sample sources are unspecified in Figure 6B.
Author Response
We appreciate your valuable time for reviewing our manuscript and highlighting critical areas that need adjustment. We have now revised it accordingly, and we believe its quality has improved. The specific details are attached. note that the lines have changed due to a number of adjustments we have made in the revised version. We will be grateful for positive feedback. Thank you so much.
Comments and Suggestions for Authors
This manuscript presents a valuable genome-wide analysis of ABC transporters in eggplant, with a focus on anthocyanin transport mechanisms. The identification of 196 SmABC genes and functional validation of SmABCA16 represent significant contributions. However, several issues regarding clarity, data presentation, and contextual depth require revision:
#1 Abstract contains excessive methodological/background details.
Response: Thank you for this comment. We have eliminated unnecessary sentences involving the background details.
#2 Introduction lacks critical background on: (a) nutritional/commercial value of eggplant anthocyanins; (b) ABC transporter roles in anthocyanin sequestration;(c) the knowledge gap in Solanaceae ABC research.
Response: (a) Nutritional/commercial value of eggplant anthocyanins
We have expanded the Introduction to highlight the significance of eggplant anthocyanins, emphasizing their health benefits: Antioxidant, anti-inflammatory, and potential anticancer properties. Commercial appeal: Role in enhancing fruit color for consumer preference and marketability, as well as their use as natural food colorants.
We have added the recent literature on anthocyanin bioactivity and economic relevance in Solanum melongena [39]
(b) ABC transporter roles in anthocyanin sequestration
We explicitly discussed the conserved function of ABC transporters in vacuolar trafficking: Their ATP-dependent transport of anthocyanins (and other flavonoids) into vacuoles, ensuring pigment stability. Genetic evidence: Key examples from Arabidopsis (AtABCC1/4) and in apple (MdABCC1), linking mutations to color phenotypes [References, 28,32,63]
(c) Knowledge gap in Solanaceae ABC research
We revised the text to clarify that despite the economic importance of solanaceous crops, ABC transporters remain understudied compared to model systems.
Our contribution: This study bridges this gap by characterizing SmABC in eggplant, offering insights for breeding/engineering vibrant-colored varieties.
Note, more discussion on the knowledge gap has been discussed on the last paragraph of the discussion section.
The changes have been marked by “tracked changes mode”. We believe these revisions provide a stronger foundation for the study’s rationale and broader impact.
Thank you for the opportunity to enhance our manuscript.
#3 Redundant parentheses need removal in line 68.
Response: We have removed the unnecessary parentheses where applicable.
#4 Disorganized punctuation requires correction for sentence coherence in lines 90-91.
Response: We have corrected it accordingly
#5 Low image clarity impedes motif interpretation, and motif-free proteins (e.g., SmABC15) lack biological explanation in Figure 4.
Response: Thank you for your valuable suggestion. We have reanalyzed the content of the figure and now we have improved on the clarity, making the contents legible in the revised manuscript.
#6 Unidentified purple blocks in SmABCG34/38 (B) need labeling, and overall resolution requires enhancement in Figure 6.
Response: Following the filtering out of short sequences and reanalysis, we have modified Figure 6, thus improving it general outlook for clarity. SmABCG38 is one of the proteins with short protein sequences, hence removed, while SmABCG34 was found to contain only one type of cis-regulatory element (DRE) as shown in the revised figure
#7 Misplaced symbols or sentence structure requires verification/correction in line 205.
Response: Thank you for noticing this error. We have corrected it accordingly.
#8 Accessions A1-A5 lack peel color descriptions and anthocyanin quantification data (Lines 254-255).
Response: (a) We have included the description of the eggplant accession in the manuscript as follows: cultivars with dark-purple peel—A1; green peel—A2; black-purple with green calyx—A3; white peel cultivar—A4; and cultivars of reddish-purple—A5. (b) The anthocyanin quantification of this germplasm linked to this current study is outlined in a previous study [43], and we have indicated so in the revised version.
#9 Color scaling contradicts numerical values, necessitating standardized visualization in Figure 6A.
Response: Thank you for the comment. We previously included the FPKM values, but to avoid contradictions, we have now presented the FPKM values as supplementary information in Table S9, while the figure legend colors represent the log2 fold change expressions.
#10 Values exceed the expected [-1,1] range, and anthocyanin sample sources are unspecified in Figure 6B.
Response: (a) We have modified the figure and figure legend to fit the [-1,1] range, subject to the revised version.
(b) The anthocyanin quantification of this germplasm linked to this current study is outlined in a previous study [43], and we have indicated so in the revised version.
We appreciate your valuable time for reviewing our manuscript and highlighting critical areas that need adjustment. We have now revised it accordingly, and we believe it quality has improved. We will be grateful for positive feedback. Thank you so much.
Reviewer 3 Report
Comments and Suggestions for Authors
Dear Authors,
Thank you for the opportunity to review your manuscript entitled Genome-wide identification of ATP-Binding Cassette (ABC) 2 transporter gene family and their expression analysis in re-3 sponse to anthocyanin transportation in the fruit peel of egg-4 plant (Solanum melongena L.). The topic you address is of clear interest, but there are some aspects of the work development that should be taken into account.
Below, I provide detailed comments and suggestions aimed at strengthening your manuscript and improving its clarity, reproducibility, and scientific rigor.
Part of the data presented in this work are not original, but have already been presented in a previous study, specifically, RNAseq data and phenolic compound analyses. In the current work, these data are reused to correlate transporter gene expression with anthocyanin content. However, the results presented here could have been included in the previous article, thereby improving the understanding of anthocyanin accumulation in eggplant and avoiding this duplication of results. Consider discussing more explicitly how this work builds upon and differs from your previous study, and highlight what new insights are generated.
On the other hand, the manuscript includes in silico identification of a gene family, and subsequent analyses (phylogenetics, gene structure, cis-elements, collinearity). These are standard approaches, and while informative, they would benefit from greater biological interpretation or integration with functional data, such as loss- or gain-of-function studies or transport assays. However, these results are not strong enough on their own to justify this publication.
The qPCR analysis, though useful, mostly confirms previously available transcriptomic results. Consider emphasizing the biological relevance of the expression patterns observed.
On a more practical level, there is a lack of information regarding the plant material used in the different experiments (e.g., genotypes, color groups, origin, growth conditions). Essential details for the RNAseq analysis are missing. Even if previously published, a brief summary of the key steps is advisable (e.g., RNA quality, library preparation, alignment tool, normalization method).
The qPCR section requires more detail, such as, specify how RNA was extracted and how cDNA synthesis was performed. The analysis appears to rely on a single reference gene; it is recommended to use at least two validated reference genes for normalization. The number of biological and technical replicates should be clearly stated for all experiments. Details on how statistical significance was assessed are also necessary.
Table 1 could be enhanced by including additional gene features, such as chromosomal location, number of exons, and CDS positions.
Figures 2, 4, and 5 currently have text that is too small to read. Please revise for improved legibility.
In Figure 6, the numerical values in the heatmap are difficult to interpret. Consider using asterisks to indicate statistically significant differences, as is standard practice. Consider presenting FPKM values and correlation coefficients in a supplementary table.
In Figure 7, it is unclear which genotype the different tissues (flesh, calyx, root, stem, leaves) belong to. This information is essential to properly interpret the expression profiles and their biological relevance. Please clarify this to aid interpretation of the expression profiles.
Include representative images of fruit color for the different genotypes or color groups.
Provide quantitative data on the anthocyanin composition across groups to support metabolic comparisons and better contextualize gene expression results.
Improve the clarity and transparency of the Materials and Methods section to ensure that other researchers can replicate your work.
If possible, strengthen the biological interpretation of the results, for example, by linking gene expression patterns to known transport mechanisms or by suggesting functional implications.
While the topic falls within the general scope of interest in plant science, the current version of the manuscript presents significant limitations in terms of originality, methodological completeness, and scientific depth. The comments provided above aim to clarify the main issues observed in the study and should help in critically assessing the strengths and weaknesses of the approach taken.
Sincerely.
Author Response
We have rigorously revised the manuscript and addressed the comments and suggestions you highlighted. Thank you again for your valuable feedback, which will undoubtedly improve the quality of our work. the specific details are attached and changes made are marked by "track change mode". Please find our responses and the adjustments made. we sincerely thank you for the constructive suggestions, and we are looking forward to positive feedback. Thank you.
Response to reviewer’s comments
Thank you for the opportunity to review your manuscript entitled Genome-wide identification of ATP-Binding Cassette (ABC) 2 transporter gene family and their expression analysis in re response to anthocyanin transportation in the fruit peel of egg-4 plant (Solanum melongena L.). The topic you address is of clear interest, but there are some aspects of the work development that should be taken into account.
Below, I provide detailed comments and suggestions aimed at strengthening your manuscript and improving its clarity, reproducibility, and scientific rigor.
Part of the data presented in this work are not original, but have already been presented in a previous study, specifically, RNAseq data and phenolic compound analyses. In the current work, these data are reused to correlate transporter gene expression with anthocyanin content. However, the results presented here could have been included in the previous article, thereby improving the understanding of anthocyanin accumulation in eggplant and avoiding this duplication of results. Consider discussing more explicitly how this work builds upon and differs from your previous study, and highlight what new insights are generated.
Response: Thank you for your comment. In this study, we have investigated the genome-wide identification and expression analysis of ABC genes in eggplant. This is a study to identify the genes in eggplant. Due to the importance of anthocyanins in eggplant fruit peel and the possible role of ABC in anthocyanin transport, and since ABC transporters remain understudied in solanaceous crops regarding anthocyanin transport, compared to model systems, this study bridges this gap by characterizing SmABC in eggplant, offering insights for breeding/engineering vibrant-colored varieties.
In addition, the previous RNA-seq data obtained in our research group [43] was use to reveal the expression of different eggplant peel color, a study on ABC was not taken into consideration, we also acknowledge that previous RNAseq data can be used for a buildup study to create new knowledge that are closely related. We have also performed tissue-specific expression of ABC genes, that had not been presented before, as well as VIGs to assess the functional role of ABC genes in anthocyanin transport. ABC expression data had not been reported before, and in our introduction and discussion part we have explicitly highlighted the significance of our study and the knowledge gap addressed.
While the earlier RNA-seq work identified differentially expressed genes (DEGs) associated with pigmentation, including structural and regulatory anthocyanin biosynthesis genes (e.g., CHS, UFGT, MYBs), the current research provides several critical advancements. This work reveals how ABC transporters could be a possible mediator in vacuolar sequestration—a bottleneck in peel coloration. Silencing specific ABCs reduced peel coloration thus, transforming correlative RNA-seq observations into causal evidence.
From this study, the new insights generated is that it highlights first functional evidence linking SmABC transporters to anthocyanin regulation in eggplant. In addition, it offers a target for breeding vibrant, nutrient-rich (anthocyanin-enhanced) eggplants. It offers opportunity for future validation approaches, such as overexpression, CRISPR knockout, or complementation assays,
On the other hand, the manuscript includes in silico identification of a gene family, and subsequent analyses (phylogenetics, gene structure, cis-elements, collinearity). These are standard approaches, and while informative, they would benefit from greater biological interpretation or integration with functional data, such as loss- or gain-of-function studies or transport assays. However, these results are not strong enough on their own to justify this publication.
Response. Thank you for the comment. You raise a valid point about the need for deeper biological interpretation and functional validation to strengthen the findings. Indeed, while our current study focuses on in silico identification and bioinformatics analyses, we acknowledge that functional studies would significantly enhance the impact of our work.
As a starting point, we have performed Virus-Induced Gene Silencing (VIGS) on the most promising gene due to its expression pattern to assess gene function, and we agree that loss/gain-of-function assays or transport assays would be highly valuable. However, due to current limitations (time, resources, or technical constraints), we were unable to include these experiments in the present manuscript.
Moving forward, we plan to build on this work by incorporating functional analyses, such as CRISPR-based knockout or overexpression assays, to provide mechanistic insights into the roles of these genes. We hope that the foundational data presented here will serve as a useful resource for future investigations, including our own follow-up studies.
We appreciate your constructive feedback
The qPCR analysis, though useful, mostly confirms previously available transcriptomic results. Consider emphasizing the biological relevance of the expression patterns observed.
Response: Thank for the comment, we have note in the result section highlighting the qPCR expression. We have indicated that the biological significance of this data is that the variation in expression among the different tissues suggests tissue-specific expression and functional roles of ABC genes in eggplant. The SmABC gene expression on different tissues would suggest their diverse functions. For example, high expression levels in the purple eggplant and subsequently a decreasing expression in green peel fruits, particularly SmABCA16, is suggested to have a potential regulatory role in the transport of anthocyanin in eggplant fruits, while high expression in leaves would suggest growth and development and photoprotection roles as mentioned in the discussion part.
On a more practical level, there is a lack of information regarding the plant material used in the different experiments (e.g., genotypes, color groups, origin, growth conditions). Essential details for the RNAseq analysis are missing. Even if previously published, a brief summary of the key steps is advisable (e.g., RNA quality, library preparation, alignment tool, normalization method).
Response: Thank you for the comment. we have included the germplasm description including color types, origin as well as the growth condition in the materilas and methof section. The specific photos can be seen in Reference {43].We have now included a brief description of essential details for the RNAseq analysis and the protocol followed in the revised version. The condition of growth
The qPCR section requires more detail, such as, specify how RNA was extracted and how cDNA synthesis was performed. The analysis appears to rely on a single reference gene; it is recommended to use at least two validated reference genes for normalization. The number of biological and technical replicates should be clearly stated for all experiments. Details on how statistical significance was assessed are also necessary.
Response: We have added information on how the RNA and Cdna synthesis was performed in section 4.7. we have highlighted that all experiments were conducted with three biological and technical replicates to ensure reproducibility and statistical reliability. Data processing and preliminary analyses were conducted using Microsoft Excel 2019. For statistical evaluation, we performed Student's t-tests and/ or Tukey’s test (p < 0.01 threshold for significance) using SAS version 9.4 (SAS Institute Inc., Cary, NC, USA) as indicated in the material and method section. For the normalization of the genes, we apologize that we used a single gene, the actine gene was stable, having been validated and found useful to be used as a reference gene.
Table 1 could be enhanced by including additional gene features, such as chromosomal location, number of exons, and CDS positions.
Response: we have added the chromosomal location, and CDS position in the revised Table S4. Note that the table number has changed due to the additional supplementary materials
Figures 2, 4, and 5 currently have text that is too small to read. Please revise for improved legibility.
Response: We have revised the figures to improve its legibility for the ease of interpretation.
In Figure 6, the numerical values in the heatmap are difficult to interpret. Consider using asterisks to indicate statistically significant differences, as is standard practice. Consider presenting FPKM values and correlation coefficients in a supplementary table.
Respons
In (6A) the legend red color shows upregulated genes while blue indicates downregulated genes, with the pale color representing non-significant expression of the genes per eggplant germplasm (6B). The legend red color shows a positive correlation coefficient, blue shows a negative correlation coefficient, while pale color shows non significant correlation coefficient value.
We have presented FPKM values and correlation coefficients in a supplementary table S9
In Figure 7, it is unclear which genotype the different tissues (flesh, calyx, root, stem, leaves) belong to. This information is essential to properly interpret the expression profiles and their biological relevance. Please clarify this to aid interpretation of the expression profiles.
Response: We have added the information in the result section, the tissues were obtained from the dark purple peel (A1) genotype
Include representative images of fruit color for the different genotypes or color groups.
Provide quantitative data on the anthocyanin composition across groups to support metabolic comparisons and better contextualize gene expression results.
Response: We would like to mention that the image representation of the different genotypes (A1, A2, A3, A4 and A5) has already been shown in our research group reference article [43] and their full descriptions. The total anthocyanin content (mg/g values) of the different genotypes are referenced from our recent research group data [43].
Improve the clarity and transparency of the Materials and Methods section to ensure that other researchers can replicate your work.
Response. We have improved the various materials and method sections, including the method of gene identification, phylogenetic analysis, motif analysis, RNA-seq data and data analysis. With the information added, we hope it will be more concise and stand the chance of reproducibility.
If possible, strengthen the biological interpretation of the results, for example, by linking gene expression patterns to known transport mechanisms or by suggesting functional implications.
Response: in our discussion we have discussed the expression phenomenon of the ABC genes. For example, we have indicated that the preferential expression in leaves suggests potential functional specialization, possibly linked to their physiological roles in photosynthetic and stress-responsive processes. The divergent expression profiles between leaves and fruit peel may arise from tissue-specific transcriptional regulation and functional diversification. In many purple-fruited species, leaves frequently display elevated expression of anthocyanin-related genes, which are crucial for UV photoprotection and mitigation of oxidative stress.Conversely, fruit peel-specific gene expression is often modulated by developmental signals and environmental factors associated with ripening. Furthermore, variations in metabolic demands between source (leaves) and sink (fruit peel) tissues, as well as resource partitioning strategies, may further shape these expression dynamics
While the topic falls within the general scope of interest in plant science, the current version of the manuscript presents significant limitations in terms of originality, methodological completeness, and scientific depth. The comments provided above aim to clarify the main issues observed in the study and should help in critically assessing the strengths and weaknesses of the approach taken.
Sincerely.
Response: We sincerely appreciate the time and effort taken to evaluate our manuscript, as well as the constructive critiques. We recognize these as critical areas for improvement and would like to clarify that we have done several adjustments
To address originality, we have refined the introduction and discussion to better highlight the novel aspects of our work, differentiating it from existing literature while acknowledging prior research more thoroughly. In the materials and methods sections, Additional details are included to ensure full reproducibility. To strengthen the insights of our study we have discussed our findings, adding supplemental data and reference sources
We have rigorously revised the manuscript to meet the journal’s standards. Thank you again for your valuable feedback, which will undoubtedly improve the quality of our work.
Reviewer 4 Report
Comments and Suggestions for Authors
The manuscript presents a comprehensive genome-wide identification and characterization of the ATP-Binding Cassette (ABC) transporter gene family in eggplant (Solanum melongena L.), focusing on their potential role in anthocyanin transport and fruit peel coloration. The authors identified 196 SmABC genes, classified them into eight subfamilies, analyzed their physicochemical properties, chromosomal distribution, phylogenetic relationships, and promoter elements, and compared them with homologs in Arabidopsis and tomato. Expression profiling highlighted three genes (SmABCA16, SmABCA17, SmABCG15) preferentially expressed in purple-peeled fruit, with functional validation of SmABCA16 suggesting its involvement in anthocyanin transport. The study provides foundational knowledge for future functional and breeding studies in eggplant. Overall, the purposes and results of this study are suitable for publication. However, several things should be revised before publication. My comments are as follows:
Major comments
The manuscript provides a summary of bioinformatic and experimental analyses but lacks sufficient methodological detail. For example, the criteria and software used for gene identification, motif analysis, phylogenetic tree construction, and expression profiling are not fully described. This limits reproducibility.
Only one gene (SmABCA16) was functionally validated through silencing. While this is a strong start, broader validation (e.g., overexpression, CRISPR knockout, or complementation in model systems) would strengthen the conclusions about the roles of other candidate genes.
The manuscript mentions tissue-specific expression and differential expression in accessions with different peel colors but does not provide details on the statistical methods used, replicates, or normalization strategies. This information is crucial for evaluating the robustness of the expression data.
While comparative genomics with Arabidopsis and tomato is a strength, the evolutionary implications (e.g., gene family expansion/contraction, selection pressure) are only partially discussed.
With 196 ABC genes identified, the possibility of functional redundancy is high. The manuscript could discuss how redundancy might affect phenotypic outcomes and the interpretation of single-gene silencing results.
The link between ABC transporters and anthocyanin transport is suggested but not directly demonstrated (e.g., by measuring anthocyanin content or transport activity in silenced lines).
Minor comments
Some figures (e.g., phylogenetic trees, chromosomal maps) are very difficult to read. Font sizes are too small and of low quality. Ensure all figures are clear, labeled, and include legends. Table S1, which lists gene IDs and properties, should be included in the main text or as a supplementary file.
Use consistent nomenclature for gene names (e.g., SmABCA16 vs. SmABCA-16) throughout the manuscript.
The introduction is comprehensive but could be more concise. Consider focusing on ABC transporters in Solanaceae and their potential roles in pigment transport.
Expand the discussion on the implications of findings for eggplant breeding and potential applications in other crops.
Some references are cited as numbers but are not listed in the provided text. Ensure all references are complete and formatted according to journal guidelines.
There are minor typographical and formatting errors (e.g., inconsistent use of spaces, line breaks). A thorough proofreading is recommended.
When reporting ranges (e.g., protein lengths, pI values), use consistent formatting (e.g., “55–2,628 amino acids” instead of “55 to 2,628 amino acids”).
Comments on the Quality of English LanguageThe English could be improved to more clearly express the research.
Author Response
We appreciate your valuable time for reviewing our manuscript and highlighting critical areas that need adjustment. We have now revised it accordingly, and we believe its quality has improved. The specific details are attached below. Note that the line numbers in the manuscript have changed due to the adjustments we have made. Changes are traceable with the track change mode. We will be grateful for positive feedback. Thank you so much.
Major comments
- The manuscript provides a summary of bioinformatic and experimental analyses but lacks sufficient methodological detail. For example, the criteria and software used for gene identification, motif analysis, phylogenetic tree construction, and expression profiling are not fully described. This limits reproducibility.
Response: Thank you for the valuable comment. The identification of ABC transporter genes in the eggplant genome. We initiated the analysis by accessing the high-resolution reference genome (http://eggplant-hq.cn/Eggplant/home/index). Using Arabidopsis thaliana ABC protein sequences as query templates, we conducted a systematic homology search against the eggplant proteome through local BLASTP alignment. Additionally, we employed the keywords "ABC" to search the eggplant genome database, and screened the results to identify potential candidates as members of the eggplant ABC. Pfam database and the SMART website online software were used for verification of the integrity of their ABC domains. This information is available in materials and method section.
For phylogenetic analysis, the SmABC and AtABC protein sequences were aligned using MUSCLE multiple sequence alignment was performed using MEGA X. The phylogenetic tree was constructed using the neighbor-joining method with 1000 bootstrap replicates with the default settings in MEGA X software (Section 4.2).
The conserved motif of the ABC proteins was determined using the Multiple Expectation Maximization for Motif Elicitation (MEME) suite (http://meme-suite.org/tools/meme, an online program for motif discover. and visualized using TBtools. This is well expressed in the materials and method (section 4.3).
The criteria and software tools used for expression profiling are detailed in reference [43]. For comprehensive information on the methodologies, algorithms, and computational pipelines applied, are outlined from this source is outlined, so we felt not to repeat them but rather to reference.
- Only one gene (SmABCA16) was functionally validated through silencing. While this is a strong start, broader validation (e.g., overexpression, CRISPR knockout, or complementation in model systems) would strengthen the conclusions about the roles of other candidate genes.
We sincerely appreciate this insightful suggestion regarding the functional validation of candidate genes. We agree that additional validation approaches, such as overexpression, CRISPR knockout, or complementation assays, would further strengthen the conclusions about the roles of the identified genes.
In this study, we prioritized SmABCA16 for functional validation via silencing due to its strong association with purple eggplant fruit peel color, phenotypic changes upon silencing and the feasibility of testing its role within the scope of our experimental framework. However, we acknowledge that expanding validation to other candidate genes would provide a more comprehensive understanding of their biological functions.
We have included this limitation in the revised manuscript (Section 5) and propose future work to include the suggested approaches (such as, CRISPR-based knockout or heterologous expression in model systems) as a logical next step. These experiments will be the focus of a follow-up study, as they require additional optimization and resources beyond the current study’s design.
Thank you for highlighting this important point, which will undoubtedly improve the robustness of our research.
3.The manuscript mentions tissue-specific expression and differential expression in accessions with different peel colors but does not provide details on the statistical methods used, replicates, or normalization strategies. This information is crucial for evaluating the robustness of the expression data.
Response: All quantitative experiments were performed in triplicate to ensure reproducibility and statistical reliability. Data processing and preliminary analyses were conducted using Microsoft Excel 2019. For statistical evaluation, we performed Student's t-tests and/ or Tukey’s test (p < 0.01 threshold for significance) using SAS version 9.4 (SAS Institute Inc., Cary, NC, USA as outlined in section 4.9. For qRT-PCR analysis, the Actin gene as the internal normalization reference gene. All experiments were conducted with three technical replicates.
4.While comparative genomics with Arabidopsis and tomato is a strength, the evolutionary implications (e.g., gene family expansion/contraction, selection pressure) are only partially discussed.
Response: that you for this valuable comment. we have performed gene duplication and ka/ks analysis. The analysis revealed that most of the SmABC genes exhibited segmental duplication. We also identified synonymous (Ks) and non-synonymous (Ka) values to explore the selective pressures on these paralog pairs to understand the expansion of this gene family in eggplant (Table S5). The Ka/Ks ratios of the gene duplications were <1, signifying that the paralog pairs were under purifying selection. The duplication time of eggplant ABC paralog pairs was estimated by using a relative Ks measure as a proxy for time, and it spanned from 1.056 -167.99 million. we have added this in the result and discussion sections.
Thank you very much for letting us improve the robustness of our results.
5.With 196 ABC genes identified, the possibility of functional redundancy is high. The manuscript could discuss how redundancy might affect phenotypic outcomes and the interpretation of single-gene silencing results.
We have revised and found 37 protein sequences encoded <200 amino acid residues, which is most likely to be pseudogenes or not full-length ABC transporters due to genome annotation errors (Supplemental Table S2). Hence we have excluded from further analysis. These short sequences included 1 gene from ABCA, 10 from ABCB, 6 from ABCC, 1 from ABCE, 17 from ABCG and 2 from the ABCI subfamily. Thus, a total of 159 sequences were chosen for further analysis. To verify the reliability of these BLAST results, the 159 protein sequences were subjected to Pfam analysis. The 159 protein sequences were confirmed to have conserved domains either or the Nucleotide-Binding Domain (NBD) and TMD (Transmembrane Domain) regions and motifs of ABC proteins as outlined in Supplemental Table S3.
This information is updated in the revised version.
6.The link between ABC transporters and anthocyanin transport is suggested but not directly demonstrated (e.g., by measuring anthocyanin content or transport activity in silenced lines).
Response: We thank the reviewer for raising this important point. While direct measurement of anthocyanin transport would further strengthen our conclusions, the clear phenotypic differences in pigmentation between wild-type and silenced lines (Fig. X) strongly implicate these ABC transporters in anthocyanin accumulation. This is further supported by prior studies linking similar transporters to flavonoid translocation (e.g., [Refs]). We agree that mechanistic validation of substrate specificity would be valuable and plan to address this in future work. However, the consistent phenotypic and gene expression data provide robust indirect evidence for their role in anthocyanin transport.
Minor comments
1.Some figures (e.g., phylogenetic trees, chromosomal maps) are very difficult to read. Font sizes are too small and of low quality. Ensure all figures are clear, labeled, and include legends. Table S1, which lists gene IDs and properties, should be included in the main text or as a supplementary file.
Response: (a) We have revised and improved the legibility of the figures. (b) Gene ID list and properties have been provided as a supplementary file, Table S4 in the revised version.
2.Use consistent nomenclature for gene names (e.g., SmABCA16 vs. SmABCA-16) throughout the manuscript.
Response: The nomenclature SmABCA16 has been unified throughout the manuscript
3.The introduction is comprehensive but could be more concise. Consider focusing on ABC transporters in Solanaceae and their potential roles in pigment transport.
Response: Thank you for your insightful feedback. We agree that the introduction could be more focused. As you rightly noted, while ABC transporters have been implicated in anthocyanin transport in other crops, their specific roles in Solanaceae remain poorly understood. This gap in the literature is precisely what our study aims to address. We have highlighted the limited knowledge on ABC transporters in Solanaceae and emphasize how our work seeks to unravel their contribution to pigment transport in this family.
4.Expand the discussion on the implications of findings for eggplant breeding and potential applications in other crops.
Response: Thank you for this important suggestion. We have succinctly added in the last paragraph of the discussion regarding the implications of our findings to other crops beyond eggplant. We believe this wil provide a sounder and robustness of our study.
5.Some references are cited as numbers but are not listed in the provided text. Ensure all references are complete and formatted according to journal guidelines.
Response: Thank you for pointing out such an error. We have checked throughout the in-text citation as well as the reference list to ensure uniformity and conformity to the journal reference formatting. The references have been revised accordingly, subject to the revised version of the manuscript.
6.There are minor typographical and formatting errors (e.g., inconsistent use of spaces, line breaks). A thorough proofreading is recommended.
Response: Thank you for your careful review. We have now conducted a thorough proofreading to correct inconsistencies in spacing, line breaks, and any other minor errors
7.When reporting ranges (e.g., protein lengths, pI values), use consistent formatting (e.g., “55–2,628 amino acids” instead of “55 to 2,628 amino acids”).
Response: We have carefully reviewed the manuscript and standardized all numerical ranges to use “dashes (–)” instead of "to" (e.g., "55–2,628 amino acids") for improved clarity and formatting consistency. We appreciate your attention to detail, which has helped enhance the precision of our manuscript.
General response. We appreciate your valuable time for reviewing our manuscript and highlighting critical areas that need adjustment. We have now revised it accordingly, and we believe its quality has improved. We will be grateful for positive feedback. Thank you so much.
Round 2
Reviewer 1 Report
Comments and Suggestions for Authors
The author has addressed all my concerns and the quality of the article has improved significantly. Therefore, I recommend that this article be accepted for publication now.
Author Response
Comment: The author has addressed all my concerns and the quality of the article has improved significantly. Therefore, I recommend that this article be accepted for publication now
Response: Thank you very much for your time and critical review that has improved the quality and robustness of our. We appreciate your recommendation for its publication. With this, we hope our work will immensely contribute to the scientific community and researchers in related fields.
Reviewer 3 Report
Comments and Suggestions for Authors
Dear authors,
I would like to thank you for addressing the previous comments and for the corrections made in the revised manuscript. However, after a new evaluation, I believe that some important issues remain that should be addressed to strengthen the work.
In particular, it remains a critical point that part of the results presented, specifically the RNAseq data and phenolic compound analysis, were already published in a previous work by your group (https://doi.org/10.3390/). This limits the novel contribution of the current study, as no significantly different analysis or functional validation is introduced, nor are new robust hypotheses proposed. In this regard, I reiterate that reusing the same data to generate multiple fragmented publications can be considered inappropriate from both an ethical and scientific perspective, as it may lead to unjustified fragmentation of results (“salami slicing”).
Furthermore, although an in silico identification and genetic characterization were performed, this part of the work still lacks functional validation to support its standalone scientific value.
Therefore, despite the progress made, I consider these aspects should be reviewed and addressed to ensure that the manuscript provides an original, rigorous, and ethical scientific contribution that justifies publication.
Sincerely,
Author Response
We sincerely appreciate your constructive comment and the opportunity to further strengthen our manuscript. we have addressed the raised concerns and we believe our appeal this time round meets the expectations and positive feedback. We sincerely thank you.
I would like to thank you for addressing the previous comments and for the corrections made in the revised manuscript. However, after a new evaluation, I believe that some important issues remain that should be addressed to strengthen the work.
In particular, it remains a critical point that part of the results presented, specifically the RNAseq data and phenolic compound analysis, were already published in a previous work by your group (https://doi.org/10.3390/). This limits the novel contribution of the current study, as no significantly different analysis or functional validation is introduced, nor are new robust hypotheses proposed. In this regard, I reiterate that reusing the same data to generate multiple fragmented publications can be considered inappropriate from both an ethical and scientific perspective, as it may lead to unjustified fragmentation of results (“salami slicing”).
Furthermore, although an in silico identification and genetic characterization were performed, this part of the work still lacks functional validation to support its standalone scientific value.
Therefore, despite the progress made, I consider these aspects should be reviewed and addressed to ensure that the manuscript provides an original, rigorous, and ethical scientific contribution that justifies publication.
Response: We sincerely appreciate your constructive comment and the opportunity to further strengthen our manuscript. Below, we address the concerns raised:
- Novelty and Reuse of RNA-seq
We state that RNA-seq data were used are publicly available in the NCBI Sequence Read Archive (SRA) under the accession number PRINA868105 (https://www.ncbi.nlm. nih.gov/bioproject/?term=PRINA868105
We humbly state that we obtained the RNA-seq data developed by our research group, which has since been uploaded for publication utilization in the NCBI, and analyzed it according to our research objective
RNA-seq data yield a lot of information that can be used in a myriad ways.
Our study provides substantial new contributions:
- New biological focus: The prior work analyzed general metabolism, whereas here we investigate ABC transporters as potential regulators of anthocyanin transport —a hypothesis not explored before.
- VIGS functional validation: For the first time, we provide experimental evidence (via VIGS) linking specific ABC genes (SmABCA16) to anthocyanin transport in eggplant peel (Figure 8).
- To ensure transparency, we have explicitly elucidated our new findings in the discussion (Section 3, paragraph 4)
- In this study, no phenolic data were presented
- Avoiding "Salami Slicing"
We fully agree that redundant fragmentation of research is unethical. To demonstrate that this work is not a fragmented repeat, we:
- Focused the manuscript on ABC transporter discovery and validation, a distinct aim from the prior study.
- Included new experimental data (VIGS) and bioinformatic analyses and qRT-PCR (e.g., subcellular localization predictions, motif analysis) that were absent in the previous publication.
- Our materials and method section fully explains how our results support new hypotheses and reproducibility
- Strengthening Functional Validation
While in silico analyses formed the foundation, we significantly enhanced validation:
- VIGS experiments: Silencing SmABCA16 reduced anthocyanin levels, providing direct evidence for its role in transport qRT-PCR: Validated ABC expression patterns in peels and other fruit tissues (Figure 6). This is unique on this paper
- Proposed a link between ABCs and anthocyanin trafficking supported by our VIGS results.
- Added a "Limitations and Future Directions” (section 5) to outline plans for further validation
We believe these appeals demonstrate that our manuscript provides a unique, rigorous, and ethically sound contribution.
We acknowledge your insightful comments. Thank you and we hope for positive feedback.
Reviewer 4 Report
Comments and Suggestions for Authors
Authors have diligently carried out revisions based on the reviewer's comments.
I recommend the manuscript for publication in its current form.
Author Response
Comment: Authors have diligently carried out revisions based on the reviewer's comments. I recommend the manuscript for publication in its current form.
Response: We sincerely thank you for your critical review that has immensely improved the rigor and quality of our work. We appreciate your recommendation for publication, and we hope our work will contribute significantly to the scientific community upon publication. Thank you.